# Single-cell imaging reveals efficient nutrient uptake and growth of microalgae darkening the Greenland Ice Sheet

Laura Halbach [1,2] ✉, Katharina Kitzinger [2,3], Martin Hansen[1,4], Sten Littmann[2], Liane G. Benning [5,6], James A. Bradley [7,8], Martin J. Whitehouse [9], Malin Olofsson [10], Rey Mourot [5,6,7], Martyn Tranter [1], Marcel M. M. Kuypers [2], Lea Ellegaard-Jensen [1] & Alexandre M. Anesio [1] ✉

Blooms of dark pigmented microalgae accelerate glacier and ice sheet melting by reducing the surface albedo. However, the role of nutrient availability in regulating algal growth on the ice remains poorly understood. Here, we investigate glacier ice algae on the Greenland Ice Sheet, providing single-cell measurements of carbon:nitrogen:phosphorus (C:N:P) ratios and assimilation rates of dissolved inorganic carbon (DIC), ammonium and nitrate following nutrient amendments. The single-cell analyses reveal high C:N and C:P atomic ratios in algal biomass as well as intracellular P storage. DIC assimilation rates are not enhanced by ammonium, nitrate, or phosphate addition. Our combined results demonstrate that glacier ice algae can optimise nutrient uptake, facilitating the potential colonization of newly exposed bare ice surfaces without the need for additional nutrient inputs. This adaptive strategy is particularly important given accelerated climate warming and the expansion of melt areas on the Greenland Ice Sheet.

The ablation zones of glacier and ice sheet surfaces are hotspots for microbial life[1–5]. Dark-pigmented glacier ice algae (*Ancylonema* spp.) are the main primary producers on bare ice surfaces[4,6], forming extensive blooms during the summer melt season[7–9]. These blooms lower the ice surface albedo, thereby accelerating ice melt[8–12]. Algal blooms on the western margin of the Greenland Ice Sheet have been shown to contribute, on average, 10 to 13% to the surface ice melt[9]. As the single largest contributor to global barystatic sea-level rise, the Greenland Ice Sheet plays a critical role in climate dynamics[13–16]. Due to climate warming, the snowline on the ice sheet is migrating to higher elevations, exposing larger areas of bare ice[17]. This expansion increases

the potential habitat area for glacier ice algae, which, if able to colonize these newly exposed ice surfaces, could accelerate ice darkening and melting. However, the triggers and controls of algal bloom progression throughout the summer ablation season and the causes of inter-annual variations in bloom extent, remain poorly understood[6,8,18–20]. Elucidating the mechanisms of algal bloom formation is crucial for predicting their future progression on bare ice surfaces[6,18,21], and for assessing their contribution to the melting of the Greenland Ice Sheet.

The highly oligotrophic conditions of the ice sheet's ablation zone may limit the growth and expansion of algal blooms on ice surfaces. Glacier surfaces are characterized by low concentrations of inorganic

¹Department of Environmental Science, iClimate, Aarhus University, Roskilde, Denmark. ²Max Planck Institute for Marine Microbiology, Bremen, Germany. ³Centre for Microbiology and Environmental Systems Science, Department of Microbiology and Ecosystem Science, University of Vienna, Vienna, Austria. ⁴Department of Environmental and Resource Engineering, Technical University of Denmark, Copenhagen, Denmark. ⁵GFZ Helmholtz Centre for Geosciences, Potsdam, Germany. ⁶Department of Earth Sciences, Freie Universität Berlin, Berlin, Germany. ⁷Aix Marseille Université, Université de Toulon, CNRS, IRD, MIO, Marseille, France. ⁸School of Biological and Behavioural Sciences, Queen Mary University of London, London, UK. ⁹Swedish Museum of Natural History, Stockholm, Sweden. ¹⁰Department of Aquatic Sciences and Assessment, Swedish University of Agricultural Sciences, Uppsala, Sweden. ✉e-mail: lhalbach@mpi-bremen.de; ama@envs.au.dk

dissolved macro-nutrients, with $NH_4^+$ and $NO_3^-$ concentrations typically below 1 μM, and aqueous $PO_4^{3-}$ below 0.1 μM[22–24]. In addition to algae, microbial communities on glacier surfaces often include protists such as ciliates and dinoflagellates, along with fungi, bacteria, and archaea[25–29], all of which compete for and drive the cycling of nutrients. McCutcheon et al.[23] found that phosphate can limit algal productivity and highlighted the positive association between phosphorus-bearing minerals, such as hydroxylapatite, and the accumulation of algal biomass. A potential relationship between minerals and algal growth has also been observed by Stibal et al.[30], who documented a positive correlation between dust loading and algal abundance.

Glacier ice algae inhabiting bare ice surfaces are likely to have evolved strategies to partially compensate for the low macro-nutrient concentrations typical of their habitats. These strategies may include maintaining a high carbon:nitrogen:phosphorus (C:N:P) biomass ratio, high nutrient uptake efficiency and/or intracellular nutrient storage capabilities. However, the nutrient content and uptake rates for cryophilic algae and, in particular, glacier ice algae, remain unknown, hindering the mechanistic understanding of potential nutrient limitations on algal growth. Previous studies have focussed solely on bulk characterization of particulate organic matter (POM) from glacier ice algal-colonised ice[23,31]. These data revealed a wide range of C:N:P ratios, spanning from 690:48:1 to 2615:196:1. The measured bulk C:P[23,31], and sometimes also bulk C:N ratios[31] in both glacier ice algae and snow algae-dominated POM samples from across the Arctic[29] were much higher than the Redfield C:N:P ratio of 116:16:1 commonly observed in marine ecosystems[32] This suggests either a relatively low macro-nutrient requirement for supraglacial algae, or potential limitations in P and N availability. However, the bulk C:N:P ratios of POM collected from ice surface samples are unlikely to accurately reflect glacier ice algae biomass stoichiometries due to elemental contributions of atmospheric deposition-derived organic matter, necromass, extracellular polymeric substances, and other microorganisms (e.g. bacteria, other eukaryotic algae, fungi) in the POM filter fraction. Hence, bulk C:N:P ratios derived from surface ice POM will invariably span a large range. Only single-cell-specific measurements can accurately determine the C:N:P ratios of glacier ice algae. Additionally, single-cell activity and nutrient uptake assessments help identify potential rate-limiting factors imposed by nutrient availability on algal growth. Thus,

targeted and cell-specific measurements are crucial for comprehending the nutrient demands that drive algal bloom progression and their future growth dynamics in glacier ecosystems.

In this study, we quantify the C:N:P ratios, assimilation rates of dissolved inorganic carbon (DIC), $NO_3^-$ and $NH_4^+$, and growth rates of single glacier ice algal cells (*Ancylonema* spp.) from the Greenland Ice Sheet both under unamended and nutrient-amendment conditions. Our aim is to gain insights into the physiological responses of this key species to varying levels of nutrient availability. In addition to single-cell analyses, we also use bulk stable isotope biogeochemical rate measurements to quantify the C and N turnover, as well as the elemental composition of the microbial community on the Greenland Ice Sheet, which we further characterize by 16S and 18S rRNA gene amplicon sequencing. Together, our findings demonstrate that glacier ice algae are well adapted to the oligotrophic conditions of the Greenland ice sheet, and exhibit no significant changes in productivity in response to external nutrient additions. This suggests that as the ablation zones of glaciers and ice sheets expand due to climate warming, newly exposed bare ice surfaces can be readily colonized by the algae without nutrient limitation hindering their growth. Our study refines the understanding of how nutrient availability influences glacier ice algal bloom development and highlights the role of algal cells in primary production and nutrient cycling on glaciers and ice sheets.

## Results

### Physico-chemical conditions on the Greenland Ice Sheet

Dark snow-free surface ice with visibly high concentrations of particulates and algal biomass was collected from the southern tip of the Greenland Ice Sheet (Fig. 1a) to assess the in situ microbial community composition and to measure bulk and single-cell elemental ratios. Incubations were performed to evaluate bulk and single-cell C-fixation and inorganic N-assimilation rates (Fig. 1b). The ice samples were allowed to melt for ~36 h at an ambient air temperature of ~4 °C, under in situ light conditions (18 h of daylight). The initial concentrations of dissolved inorganic nutrients in the melted ice were 0.08 μM $NH_4^+$, 0.05 μM $NO_3^-$, and ~0.01 μM $PO_4^{3-}$ (Table 1). Dissolved organic nitrogen and phosphorous (DON and DOP) were present at concentrations ~5 and ~7 times higher than their respective inorganic counterparts.

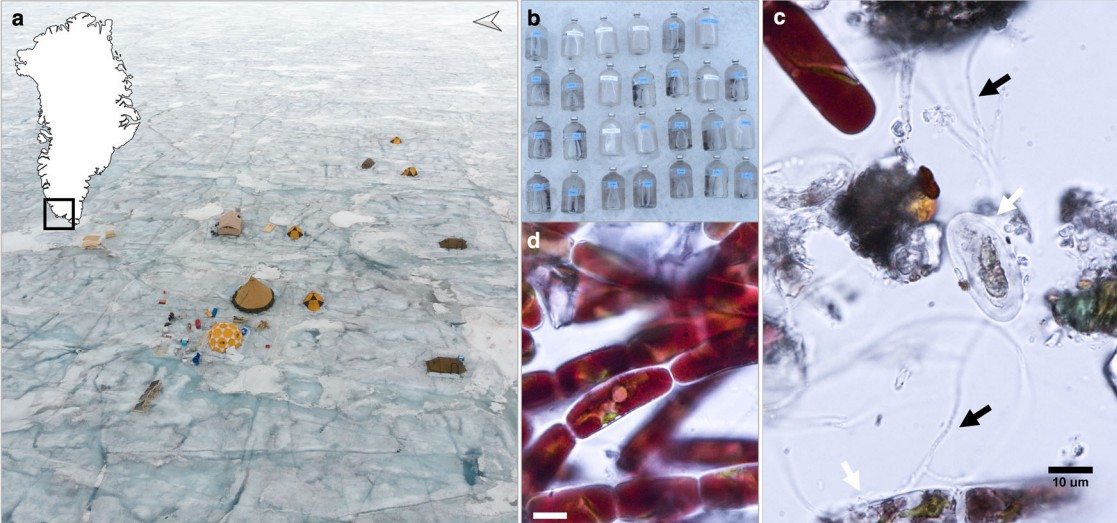

**Fig. 1 | Overview of the sampling site and experimental setup on the Greenland Ice Sheet. a** The black square in the map insert shows the location of the sampling site, located close to the PROMICE station QAS_M, upwind and next to the ice camp to minimize potential contamination. **b** Incubation of unfiltered melted ice samples in replicate serum bottles on the surface of the ice sheet. **c** Microscopic image of the supraglacial community prior incubation. Black arrows indicate fungal hyphae, and white arrows point to putatively dead *Ancylonema nordenskiöldii* cells, potentially infected by parasitic fungi. **d** Microscopic image of the supraglacial glacier ice algae, *Ancylonema nordenskiöldii*. Scale bars in (**c**) and (**d**) are 10 μm.

**Table 1 | Nutrient and base cation concentrations, along with the composition of the algal community in the initial surface meltwater sample from the Greenland Ice Sheet (prior to incubation)**

| Parameter | | Value | Range | n |
|---|---|---|---|---|
| Nutrient concentrations in situ (µM) | $NH_4^+$ | 0.08 | | 1 |
| | $NO_3^-$ | 0.05 | | 1 |
| | $NO_2^-$ | <0.01 | | 1 |
| | $PO_4^{3-}$ | ~0.01 (<LOQ of 0.02) | | 1 |
| | DON | 0.65 | | 1 |
| | DOP | 0.1 | | 1 |
| | $Ca^{2+}$ | 0.66 | | 1 |
| | $Mg^{2+}$ | 0.36 | | 1 |
| | $Na^+$ | 1.54 | | 1 |
| | $K^+$ | 0.22 | | 1 |
| Community composition in situ | A. nordenskiöldii filaments (filaments ml⁻¹) | 3880 | 3770–3990 | 2 |
| | A. nordenskiöldii chain length (number of cells) | ~3 | 1–18 | 2 |
| | A. nordenskiöldii abundance (cells ml⁻¹) | 10,700 | 10,480–10,890 | 2 |
| | A. alaskanum abundance (cells ml⁻¹) | 5580 | 4500–6650 | 2 |
| | Abundance snow algae (cells ml⁻¹) | 2110 | 1970–2260 | 2 |

Values are presented as means (except for *A. nordenskiöldii* filament length, reported as median), with corresponding ranges and sample sizes (*n*).

## Supraglacial community composition

The bare ice microbial community sampled for the incubation experiments had a mean glacier ice algal (phylum *Phragmoplastophyta*) abundance of $16.2 \pm 1.2 \times 10^3$ cells ml⁻¹ based on microscopic analyses, of which ~66% were filamentous *Ancylonema* cf. *nordenskiöldii* and ~34% were unicellular *A.* cf. *alaskanum* (Table 1). Red-coloured snow algal cysts (phylum *Chlorophyta*), *Chlamydomonas* spp., were present at an abundance of $2.1 \pm 0.2 \times 10^3$ cells ml⁻¹, and represented 12% of the total eukaryotic algal cells. In total, 76 amplicon sequence variants (ASVs) were found within the 18S rRNA gene amplicon data, confirming the dominance of glacier ice algae within the eukaryotic community. The phylum *Phragmoplastophyta* had the highest relative abundance among the eukaryotes (70%; Supplementary Fig. 1a) and was comprised solely of *Mesotaeniaceae*. Fungi contributed ~19% of the eukaryotic community, with the phyla *Basidiomycota*, *Ascomycota*, and *Chytridiomycota* dominating. Microscopic observations confirmed the presence of pigmented glacier ice algae with parasitic infections by *Chytridiomycota* (Supplementary Fig. 2). Other eukaryotic phyla found at lower relative abundances were *Chlorophyta* (9%), *Cercozoa* (1%), and *Ciliophora* (<1%). The bacterial community (determined by 16S rRNA gene amplicon sequencing) was dominated by *Bacteroidia* (37%), followed by *Actinobacteria* (26%), and *Alphaproteobacteria* (14%). *Cyanobacteria* represented 4% of the bacterial community. Overall, 71 bacterial ASVs were found.

## In situ glacier ice algae C:N:P ratios

Elemental mapping of single glacier ice algal cells in the fresh ice melt yielded a mean in situ C:N biomass ratio of $19 \pm 2.9$ and a C:P ratio of $509 \pm 149$, exceeding the Redfield C:N (6.6) and C:P ratio (116) four- and three-fold, respectively[32] (Fig. 2; Table 2). The mean glacier algal N:P ratio was $26 \pm 5$, higher than the Redfield N:P stoichiometry of 16. Overall, we observed a high variability in cellular C:P, N:P, and C:N stoichiometries. Notably, elemental mapping also revealed the presence of small (<1 µm) P-rich inclusions inside individual glacier ice algae cells (Fig. 2, white arrows in P elemental map).

## C-fixation, N-assimilation, and the effect of nutrient additions in bulk samples and single algal cells

Incubations were performed under in situ conditions on the ice surface (Fig. 1b) to quantify the activity of both single glacier ice algae cells (Figs. 3; 4d, e; Table 2) and the bulk microbial community

(Fig. 4a, b; Table 2) using stable isotope-based measurements of DIC and DIN uptake. All incubations received $^{13}C$-DIC to assess photoautotrophic C fixation, either with no nutrient amendments (control), or addition of $^{15}N$-$NH_4^+$, $^{15}N$-$NO_3^-$, $PO_4^{3-}$ or combined $^{15}N$-$NH_4^+ + PO_4^{3-}$. Following isotope amendments, samples were taken after 6 h (T1) and 30 h (T2) of incubation. The incorporation rates of the C or N isotopes were calculated based on the change in isotopic composition of biomass from algal cells and bulk POM over the incubation period. The DIC concentration measured at T1 from the control bottle was 275 µM, much higher than 44–70 µM reported in Andrews et al.[33] or 15 µM observed in Yallop et al.[12], with the latter closer to values expected in dilute glacier ice melt in equilibrium with the atmosphere (Supplementary Note 1). The high initial DIC concentration might suggest that net heterotrophic activity and/or photooxidation occurred during the incubations and was likely impacted by the process of ice melting, but these values mitigate potential DIC limitations during the incubations.

Bulk and single-cell isotope incorporation measurements were obtained for T1 and T2 of the incubation period, except for single-cell analyses, where T1 measurements were only performed for the control and $^{15}N$-$NH_4^+$ treatments. In total, 244 glacier algal cells were analysed using high-resolution secondary ion mass spectrometry (HR-SIMS), with 24 of the 244 imaged cells (~10%) not exhibiting any DIC uptake (Fig. 4d, e). The growth and assimilation data of algal cells described here and in Table 2 are based solely on the active fraction of the population (those showing DIC assimilation), while a comprehensive overview of the HR-SIMS data, including inactive cells, is provided in Supplementary Table 1.

Nutrient additions did not stimulate increases in bulk or single-cell C-based growth rates. Rather, the $^{15}N$-$NH_4^+$, $^{15}N$-$NO_3^-$, and combined $^{15}N$-$NH_4^+ + PO_4^{3-}$ additions resulted in significantly decreased bulk C-based growth rates (growth rates of $0.36 \pm 0.03$, $0.29 \pm 0.05$, $0.24 \pm 0.04$ day⁻¹, respectively) compared to the control ($0.63 \pm 0.03$ day⁻¹; Kruskal-Wallis: chi-squared = 12, $p = 0.02$, df = 4; Fig. 4a). Similarly, the mean C-based growth of single glacier ice algal cells also did not show any stimulation upon nutrient addition, but their C-based growth was significantly decreased in the $PO_4^{3-}$ and the $^{15}N$-$NH_4^+ + PO_4^{3-}$ treatments ($0.20 \pm 0.11$, $0.23 \pm 0.16$ day⁻¹, respectively; Kruskal-Wallis: chi-squared = 47, $p = 1.56e^{-9}$, df = 4) compared to the control ($0.47 \pm 0.24$ day⁻¹) (Fig. 4d). The mean C-based growth rates for

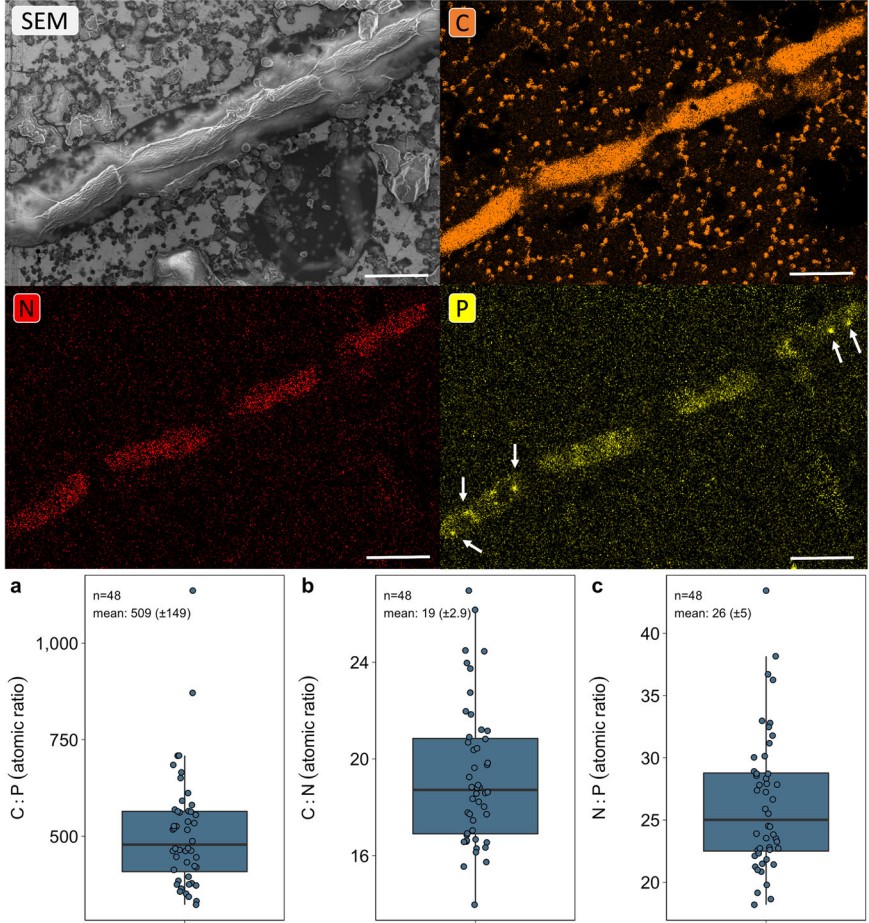

**Fig. 2 | Elemental content of glacier ice algae analysed by scanning electron microscopy (SEM) combined with energy-dispersive X-ray spectroscopy (EDS).** Image shows a representative SEM image (upper left panel) and corresponding elemental maps for C (upper right panel), N (lower left panel) and P (lower right panel). Note the presence of P-rich granules within some of the glacier ice algal cells (white arrows, lower right panel). Scale bars are 10 μm. **a–c** In situ (prior to incubation, T0) atomic ratios of C, N and P in single glacier algal cells ($n$ = 48 cells), with boxplots showing the 25–75% quantile range, the median as a line, and whiskers extending to 1.5x the interquartile range of the data. Source data are provided as a Source Data file.

single glacier ice algal cells in the control treatment, with values of $0.65 \pm 0.36$ at T1 and $0.47 \pm 0.24$ day$^{-1}$ at T2, correspond to mean C-based doubling times of ~2 days (Fig. 4d; Table 2).

The bulk and single-cell N-based growth rates were all higher at T1 compared to T2, which is consistent with the rapid depletion of the N-tracers in solution: <37% of NO$_3^-$ in the $^{15}$N-NO$_3^-$ treatment and <3% of the NH$_4^+$ in the $^{15}$N-NH$_4^+$ treatment remained after the first 6 h of incubation (T1) (Supplementary Fig. 4a, b). There was no statistically significant difference in N-based growth rates between the $^{15}$N-NH$_4^+$, $^{15}$N-NO$_3^-$, and $^{15}$N-NH$_4^+$ + PO$_4^{3-}$ additions in both bulk and single-cell measurements when compared at the same timepoint (Kruskal-Wallis: chi-squared = 4.7, df = 3, $p$ = 0.2 and Kruskal-Wallis: chi-squared = 0.63, df = 2, $p$ = 0.7) (Fig. 4b, e). The bulk N-based growth rates for both NO$_3^-$ and NH$_4^+$ were 0.60 day$^{-1}$ at T1, with assimilations of 44.8 and 32.4 μmol N L$^{-1}$ day$^{-1}$, respectively (difference in values is likely due to inhomogeneous biomass distribution; assimilation normalised to biomass: 28.5 and 28.6 μmol N mg$^{-1}$ N$_{POM}$ day$^{-1}$, respectively) (Fig. 4b, Table 2). The bulk N-based growth rates were 0.2 day$^{-1}$ for T2 for both NO$_3^-$ and NH$_4^+$. The N-based growth rates of the $^{15}$N-NH$_4^+$ and $^{15}$N-NO$_3^-$ treatments were similar also for single glacier ice algal cells, at 0.07 day$^{-1}$ at T2 (Fig. 4e, Table 2). The bulk POC:PON ratio in the nitrogen-spiked treatments decreased between T0 and T1 compared to the control and consistently remained lower, in line with the rapid depletion of the nitrogen tracers in solution (Fig. 4c).

Glacier ice algal cells assimilated C in excess of N and generally in excess of the Redfield C:N ratio (Fig. 5). C-fixation of the single algal cells continued at high rates, despite the measurable N-assimilation slowing down (due to the depletion of the $^{15}$N-tracers) at later time points, with mean C:N assimilation ratios of ~46 at T$_1$ and ~84–113 at T2. There was no significant correlation between cell volume and C-based growth rates (Supplementary Fig. 5). Overall, we observed a high variability in C-fixation and N-assimilation rates between glacier ice algal cells (Figs. 4d, e; 5).

### Contribution of glacier ice algae to bulk C and N uptake

Using the single-cell rate measurements, alongside bulk C and N uptake rates from DIC or NH$_4^+$, and the algal abundance, we estimate the glacier ice algae contribution to the total C and N uptake in bulk POM. These calculations are sensitive to variability in biomass distribution among incubation bottles, differences in single-cell activity rates, and variability in algal cell abundance between bottles and timepoints (Supplementary Note 2). The bulk POM measurements encompass all particulate matter retained on filters (3 and 0.2 μm), including not only glacier ice algae but also other organisms, such as dispersed cryoconite material containing cyanobacteria and organic matter (SEM images in Supplementary Fig. 6). Glacier ice algal assimilation accounted for ~7 ± 6% to 15 ± 12%, of the $^{13}$C from DIC recovered in POM, while glacier ice algae accounted between ~3 ± 2% to 8 ± 6% of the $^{15}$N from NH$_4^+$ recovered in POM (Supplementary Tables 2 and 3).

**Table 2 | Bulk and single-cell elemental compositions and activity rates of a surface ice community on the Greenland Ice Sheet. Data are shown as means with standard deviations (SD)**

| Parameter | | Mean | | SD | n |
|---|---|---|---|---|---|
| Bulk particulate organic C and N contents in situ | POC (µmol C L$^{-1}$) | 2849$_{(T0)}$ | ± | 1015$_{(T0)}$ | 3 |
| | PON (µmol N L$^{-1}$) | 142$_{(T0)}$ | ± | 56$_{(T0)}$ | 3 |
| Bulk particulate C and N uptake and growth | C assimilation (µmol C L$^{-1}$ day$^{-1}$)$^a$ | 448$_{(T2)}$–1879$_{(T1)}$ | ± | 101$_{(T2)}$ | 4 |
| | C assimilation (µmol C mg$^{-1}$ C$_{POM}$ day$^{-1}$)$^a$ | 28.2$_{(T2)}$–33.9$_{(T1)}$ | ± | 1.68$_{(T2)}$ | 4 |
| | C-based growth rate (day$^{-1}$)$^a$ | 0.62$_{(T1)}$–0.63$_{(T2)}$ | ± | 0.03$_{(T2)}$ | 4 |
| | C-based doubling time (days)$^a$ | 1.6$_{(T1)}$–1.63$_{(T2)}$ | ± | 0.09$_{(T2)}$ | 4 |
| | NH$_4^+$ assimilation (µmol N L$^{-1}$ day$^{-1}$)$^b$ | 9.07$_{(T2)}$–32.4$_{(T1)}$ | ± | 1.03$_{(T2)}$ | 4 |
| | NO$_3^-$ assimilation (µmol N L$^{-1}$ day$^{-1}$)$^b$ | 9.59$_{(T2)}$–44.8$_{(T1)}$ | ± | 2.40$_{(T2)}$ | 4 |
| | NH$_4^+$ assimilation (µmol N mg$^{-1}$ N$_{POM}$ day$^{-1}$)$^b$ | 8.22$_{(T2)}$–28.6$_{(T1)}$ | ± | 1.41$_{(T2)}$ | 4 |
| | NO$_3^-$ assimilation (µmol N mg$^{-1}$ N$_{POM}$ day$^{-1}$)$^b$ | 8.0$_{(T2)}$–28.5$_{(T1)}$ | ± | 1.35$_{(T2)}$ | 4 |
| | N-based growth rate (NH$_4^+$) (day$^{-1}$)$^b$ | 0.18$_{(T2)}$–0.60$_{(T1)}$ | ± | 0.03$_{(T2)}$ | 4 |
| | N-based growth rate (NO$_3^-$)(day$^{-1}$)$^b$ | 0.17$_{(T2)}$–0.61$_{(T1)}$ | ± | 0.03$_{(T2)}$ | 4 |
| Single-cell elemental composition in situ | C:N atomic ratio | 19$_{(T0)}$ | ± | 2.9$_{(T0)}$ | 48 |
| | C:P atomic ratio | 509$_{(T0)}$ | ± | 149$_{(T0)}$ | 48 |
| | N:P atomic ratio | 26$_{(T0)}$ | ± | 5$_{(T0)}$ | 48 |
| Single-cell C and N assimilation and growth | $^{13}$C assimilation (DIC) (pmol C cell$^{-1}$ day$^{-1}$)$^a$ | 4.6$_{(T2)}$–9.7$_{(T1)}$ | ± | 3.46$_{(T2)}$–5.5$_{(T1)}$ | 48 |
| (active population) | $^{15}$N assimilation (NH$_4^+$) (fmol N cell$^{-1}$ day$^{-1}$)$^b$ | 37.7$_{(T2)}$–183$_{(T1)}$ | ± | 23.2$_{(T2)}$–137$_{(T1)}$ | 45 |
| | $^{15}$N assimilation (NO$_3^-$) (fmol N cell$^{-1}$ day$^{-1}$)$^b$ | 63.5$_{(T2)}$ | ± | 64.1$_{(T2)}$ | 32 |
| | $^{13}$C-based growth (DIC) (day$^{-1}$)$^a$ | 0.47$_{(T2)}$–0.66$_{(T1)}$ | ± | 0.24$_{(T2)}$–0.37$_{(T1)}$ | 48 |
| | $^{15}$N-based growth (NH$_4^+$) (day$^{-1}$)$^b$ | 0.07$_{(T2)}$–0.18$_{(T1)}$ | ± | 0.03$_{(T2)}$–0.05$_{(T1)}$ | 45 |
| | $^{15}$N-based growth (NO$_3^-$) (day$^{-1}$)$^b$ | 0.07$_{(T2)}$ | ± | 0.03$_{(T2)}$ | 32 |
| | $^{13}$C-derived population doubling (days) (DIC)$^a$ | 2.12$_{(T2)}$–1.53$_{(T1)}$ | ± | 1.06$_{(T2)}$–0.84$_{(T1)}$ | 48 |
| | $^{15}$N-derived population doubling (days) (NH$_4^+$)$^b$ | 14.3$_{(T2)}$–5.55$_{(T1)}$ | ± | 5.9$_{(T2)}$–1.7$_{(T1)}$ | 45 |

The number of replicates (n) for the bulk parameter correspond to the analysed replicate samples and for the single-cell data to the number of analysed cells (n). Single-cell assimilation and growth data represent estimates of the active fraction of the algal population and the estimates for single-cells from T2 are derived from one replicate bottle of the respective treatment. Ranges in $^{13}$C-DIC, $^{15}$N-NO$_3^-$ and $^{15}$N-NH$_4^+$ assimilation rates correspond to different incubation lengths (T1 or T2). For single-cell measurements of the entire population (active and inactive cells), see Supplementary Table 1.
$^a$Measurements derived from the control treatment.
$^b$Measurements derived from either the $^{15}$NH$_4^+$ or $^{15}$NO$_3^-$-amended treatment.

## Discussion

We studied the activity, nutrient storage and uptake, and growth response to nutrient availability of the microbial population at the southern margin of the Greenland Ice Sheet surface. Our study documented how variability in stoichiometric ratios and uptake of C and nutrients in single glacier ice algae cells compare to bulk measurements of POM in surface ice samples. We found an active glacier ice algal community, with a mean C-based doubling time of -2±4 days (Table 2), which is comparable to previously measured primary production-based doubling times on the Greenland Ice Sheet[8]. The C-based growth and C-assimilation rates for single cells of glacier ice algae are of the same order of magnitude as those of autotrophic marine diatoms or dinoflagellates[34,35]. Our nutrient-addition experiments demonstrated that increased concentrations of $^{15}$N-NH$_4^+$, $^{15}$N-NO$_3^-$, PO$_4^{3-}$, and $^{15}$N-NH$_4^+$ + PO$_4^{3-}$ did not enhance C-based growth of the bulk community or single algal cells over a 30-hour incubation period (Fig. 4). In fact, we observed that C-based growth was equal to or even lower under the tested nutrient-loading scenarios. We interpret this finding to be indicative of sufficient nutrient availability under the conditions at the time of sampling (Table 1). Concentrations of nutrients in our samples were comparable to or lower than those reported in other areas of the Greenland Ice Sheet[24]. These findings, which show no evidence of nutrient limitation for glacier ice algae, are consistent with recent work by Feng et al.[36] identifying the duration of bare ice exposure as the primary driver of ice darkening. While Feng et al.'s study focuses on the combined effects of all light-absorbing particles and meltwater accumulation on ice darkening, our results suggest that, for algae, the absence of nutrient limitation

further underscores the importance of the duration of bare ice exposure as a key factor in promoting the development of algal blooms and subsequent ice darkening. The absence of nutrient stimulation effects on glacier ice algal productivity aligns with the findings by McCutcheon et al.[23] where, although phosphorus was proposed as a major control on algal growth, the increase in maximum rates of electron transport (a proxy for photosynthesis) upon PO$_4^{3-}$ additions occurred only after 5 days of incubation. The effect of PO$_4^{3-}$ addition was, hence, only evident while incubating in a closed system and without ongoing nutrient supply, for instance by surface ice ablation[37] or atmospheric deposition[38–41]. In contrast, the shorter incubation times carried out in this study minimised potential bottle effects and substrate transfer (cross-feeding) between microorganisms. Nevertheless, the potential impacts of closed-bottle incubations and high nutrient loads on microbial metabolism cannot be ruled out. The slower C-based growth under the nutrient-loading scenarios could reflect a sensitivity of the microbial community to high nutrient loads, as they may be adapted to the highly oligotrophic conditions of glacial surfaces[42]. While trace elements such as iron, zinc, manganese, and cobalt were not analysed in this study, it is important to note that these metals are essential cofactors for enzymes involved in central cellular processes such as photosynthesis and nutrient assimilation[43]. Trace metal data from algae-dominated, oligotrophic supraglacial environments are emerging, with early indications pointing to a potential link between low summer macronutrient and trace metal concentrations and the occurrence of algal blooms[20,29]. Trace metal availability and micronutrient requirements of glacier ice algae, along with other potential growth

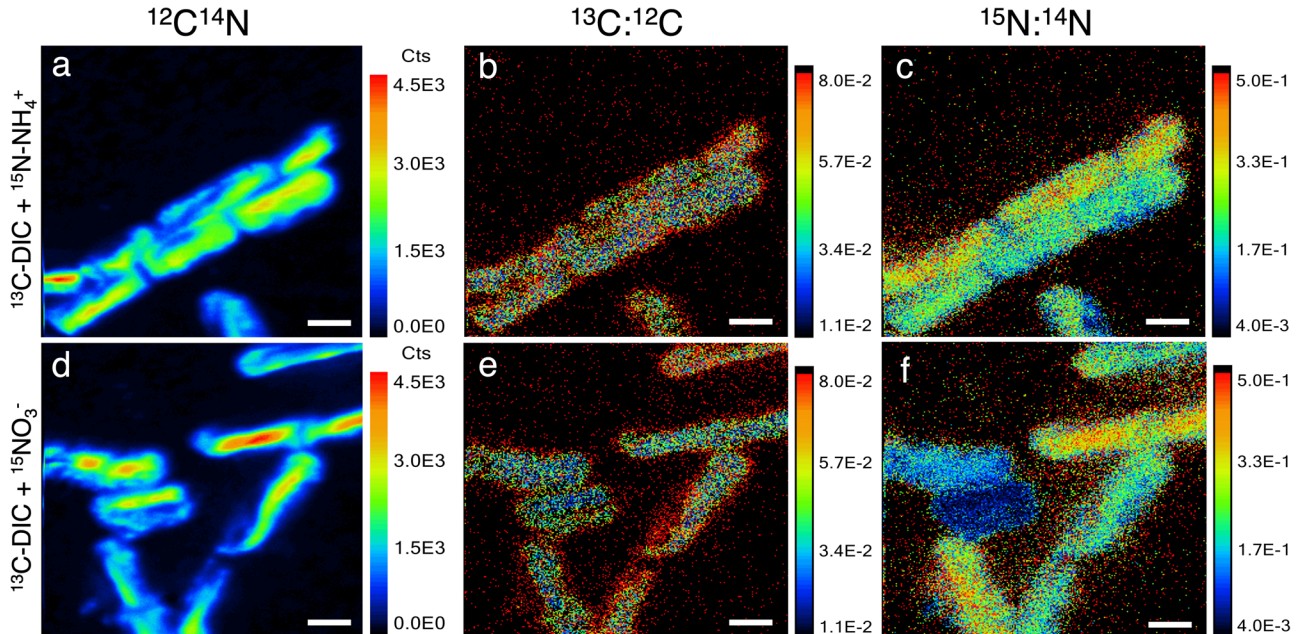

**Fig. 3 | High-resolution secondary ion mass spectrometry (HR-SIMS) imaging of glacier ice algal cells following 30-h incubations with $^{13}$C-DIC and $^{15}$N tracers.** **a–c** Incubations with $^{13}$C-DIC and $^{15}$N-NH$_4^+$. **d–f** Incubations with $^{13}$C-DIC and $^{15}$N-NO$_3^-$. **a, d** $^{12}$C$^{14}$N ion counts per pixel, a proxy for algal biomass. **b, e** $^{13}$C:$^{12}$C ratio, a proxy for DIC assimilation; and (**c, f**) $^{15}$N:$^{14}$N ratio, a proxy for NH$_4^+$ and NO$_3^-$ assimilation. Note the heterogeneity in both $^{13}$C and $^{15}$N enrichments between individual cells. Scale bars are 10 μm. Source data are provided as a Source Data file and data is shown in Fig. 4d, e.

constraints, such as infections by parasitic fungi[25,44,45] or viruses[28,46], require further investigation, particularly in the context of the ongoing prolongation of bare ice exposure. We demonstrated a rapid uptake of inorganic N sources by the microbial community, occurring within just a few hours. This was reflected by the decreasing NH$_4^+$ and NO$_3^-$ concentrations during incubations (Supplementary Fig. 4a, b), a decrease of POC:PON ratio in the treatments receiving $^{15}$N-NH$_4^+$ and $^{15}$N-NO$_3^-$ relative to the control (Fig. 4c), and substantial $^{15}$N labelling of algal cells in $^{15}$N-NH$_4^+$ and $^{15}$N-NO$_3^-$ amended incubations (Figs. 3, 4e). Rapid inorganic N assimilation by algal cells is typical of organisms in oligotrophic systems[47] and likely reflects the ability of glacier ice algae to maximise nutrient uptake and store excess N when available, even when N is not limiting. Indeed, low in situ nutrient concentrations are often associated with high turnover rates in aquatic environments[48]. Consistent with microbial utilization and recycling of N, DON concentrations increased by a factor of 3 in the N-amended treatments compared to the control treatment during the first 6 h of incubation (Supplementary Fig. 4c). In addition to possible N-storage, our SEM elemental data showed that glacier ice algae store phosphorus granules (Fig. 2, P elemental map), possibly as polyphosphate, a common biological phosphorus storage mechanism in algae and all other domains of life[49–53]. Such P storage abilities have been reported, for example, in Arctic *Cylindrocystis* strains (Zygnematophyceae)[54]. Excess storage of nutrients may sustain the metabolic requirements and possibly even allow for the growth of glacier ice algae towards the end of the melt season when nutrient availability may be scarcer.

We show that the low cellular N and P content relative to C reflects the overall low nutrient requirements of glacier ice algae, rather than indicating nutrient limitation. This is demonstrated by the observation that the algal cells that had a relatively high in situ C:N and C:P biomass content (mean of 509:26:1; Fig. 2) did not show signs of nutrient limitation during the short-term incubations upon nutrient addition (Fig. 4d, e). The average stoichiometric ratios of glacier ice algae clearly exceed the marine-derived Redfield ratio (C:N = 6.6, C:P of 116:1) and reflect the very different growth conditions in the dilute, oligotrophic

melting ice habitat, which requires markedly different physiological adaptations for glacier ice algae relative to marine eukaryotic autotrophs. During the summer ablation season, with high light intensity, low nutrient availability, and continuous DIC supply[31], photosynthesis can become decoupled from growth, resulting in excess production of photosynthate that cannot be used for growth[55–58]. The increasing DIC:DIN assimilation ratio that we observed (Fig. 5) likely reflects this decoupling of primary productivity and DIN uptake in the algal cells. Excess photosynthate may be allocated to C-rich, N and/or P-poor compounds, such as storage lipids, carbohydrates[55–57,59,60], or phenolic pigments[31,61], all of which contribute to an increase in cellular C content, or it may be excreted as sugars. Notably, phenolic pigments accumulate in the cells in large quantities—more than 20 times higher than Chlorophyll $a$[21,62]—helping to protect the algae from damagingly high UV irradiance, dissipate heat, and act as sinks for excess photosynthate (likely contributing to the elevated C content in the cells). We reason that the glacier ice algae exhibit an elevated C:N and C:P content compared to the Redfield ratio due to their distinct physiological adaptations to the supraglacial habitat. The deviation from the Redfield ratio that we observed is consistent with previous assessments (Williamson et al.[31] and Lutz et al.[29]) using bulk material (POM). It is important to note, however, that in contrast to the single-cell analysis conducted here, bulk POM may also capture other organisms (e.g., other eukaryotes, fungal biomass, dispersed cryoconite with cyanobacteria), partially degraded necromass, and organic matter. Thus, the single-cell stoichiometric measurements provide a more direct representation of the elemental ratios of glacier ice algae and offer insights into their nutrient retention processes.

We suggest that freshly fixed C is being transferred from primary producers to the heterotrophic community via cross-feeding. This is supported by the estimated contribution of active algal cells, of only ~7 ± 6 to 15 ± 12%, to the total bulk C uptake recovered in bulk POM from $^{13}$C-DIC. Although algal biomass was heterogeneously distributed between incubation bottles and the variability in single-cell activity was large, the estimated upper limit of the contribution of active algal biomass to bulk C uptake is small and gives insights into C cycling

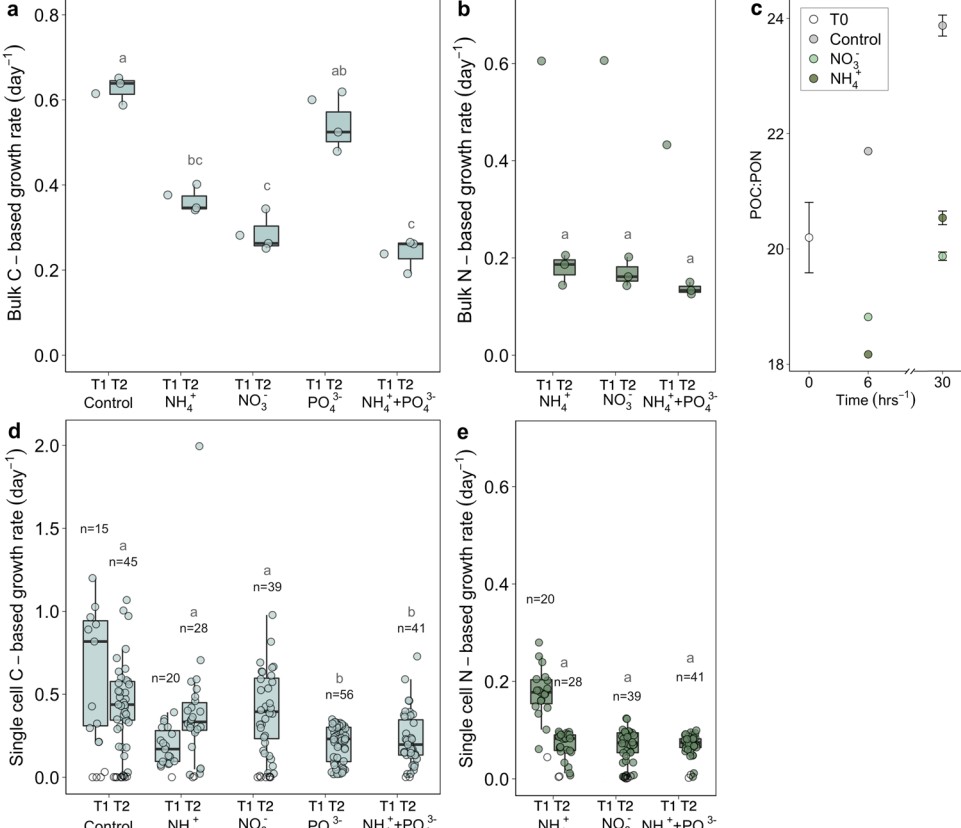

**Fig. 4 | Growth rates of the bulk community, changes in POC:PON ratios and growth rates of single glacier ice algal cells.** Growth rates based on (**a**) $^{13}$C-DIC and (**b**) $^{15}$N-NH$_4^+$ or $^{15}$N-NO$_3^-$ incorporation for the bulk community under different nutrient treatments at T1 (calculated as change from T0 to T1; 6 h of incubation, $n = 1$) and T2 (calculated as change from T0 to T2; 30 hrs of incubation, $n = 3$) with n representing the number measured bottles. **c** Changes in POC:PON ratios for control, $^{15}$N-NH$_4^+$, and $^{15}$N-NO$_3^-$ treatments at T1 ($n = 1$ measured bottle) and mean ratios at T2 ($n = 3$ measured bottles) relative to the average in situ ratio at T0 ($n = 3$ replicate samples). Error bars represent the standard error. Growth rates based on

(**d**) $^{13}$C-DIC and (**e**) $^{15}$N-NH$_4^+$ or $^{15}$N-NO$_3^-$ incorporation of individual glacier ice algal cells (**d**: $n = 244$ cells, **e**: $n = 128$ cells), with inactive cells marked as white dots. **a**, **b**, **d**, **e** Statistically significant differences between treatments at T2 are indicated by different lower-case letters: (**a**) Kruskal-Wallis, chi-squared = 12, $p = 0.02$, df = 4. **b** Kruskal-Wallis, chi-squared = 4.7, df = 3, $p = 0.2$. **d** Kruskal-Wallis, chi-squared = 47, $p = 1.56e^{-9}$, df = 4. **e** Kruskal-Wallis, chi-squared = 0.63, df = 2, $p = 0.7$. Boxplots show the 25–75% quantile range, the median as a line, and whiskers extending to 1.5× the interquartile range. Source data are provided as a Source Data file.

within glacier microbial food webs. Other autotrophic taxa comprised only a minor fraction of the total autotrophic community (cyanobacteria: 4% of bacterial ASVs; snow algae: ~10% of algal counts), and it is, therefore, likely that glacier ice algae were the main primary producers in our incubations. We thus deduce that a large fraction of the DIC assimilated by glacier ice algae was rapidly released as DOC (e.g., as exopolymeric substances that can stick to POM on the filters[23]) and could be assimilated by the microbial community. We hypothesize that some freshly fixed C from glacier ice algae was also transferred to *Chytridiomycota*, as these parasitic fungi rely entirely on autotrophic C from host cells[63]. We detected a high relative abundance of *Chytridiomycota* among the eukaryotic ASVs in our samples (Supplementary Fig. 1a). Microscopy further showed the presence of fungal hyphae (Fig. 1c) and numerous algal cells with signs of parasitic infections (Supplementary Fig. 2). We, therefore, reason that glacier ice algae may be an important source of rapidly available organic C to the microbial food web on glacier surfaces.

Our combined single-cell elemental and isotopic imaging revealed that the elemental composition of glacier ice algae and their C- and N-based growth rates were highly variable (Figs. 3; 4d, e; 5). Phenotypic intrapopulation variability was also evident microscopically, with individual cells differing in size, cell division stage or pigmentation (Fig. 1c, Supplementary Fig. 2). The striking variability in DIC and DIN uptake rates among the algal cells may explain some of the observed variability in the single-cell elemental ratios. This

variability, along with differences in activity modes, growth stages[64], selective allocation of C, N or P to cell maintenance (including biomolecule replacements and/or repair)[65,66], and micro-scale variations in nutrient availability, may all influence the C:N:P ratio of individual cells. Intrapopulation variability of microorganisms is ubiquitous in nature[67], and variable activity and phenotypic traits within glacier algal populations are likely crucial for adapting to environmental gradients, including resource availability, both over time and spatially at the microscale within the ice biofilm. From our measurements, we also provide the assessment of the active/alive (90%) and non-active/dead (10%) fractions of the glacier ice algae population (Fig. 4d). The fraction of active cells may depend on factors such as season, location, rate of dead cell degradation, and infections by parasitic fungi *Chytridiomycota*[25,44,45] or viruses[28,46]. Overall, single-cell analyses provide key insights into the adaptive responses and microbial interactions of microbes.

As our climate warms, the Greenland Ice Sheet faces prolonged summer ice melt, which may extend the duration and magnitude of algal blooms on its surface. New bare ice surfaces may also be colonized by algae if sufficient nutrients are available to support their growth[23,24,31]. However, direct measurements of nutrient uptake and growth in glacial microbial communities have thus far been lacking, limiting our understanding of their nutrient requirements in these oligotrophic glacier environments. We address this gap with measurements of dual DIC and NH$_4^+$ or NO$_3^-$ assimilation, as well as the

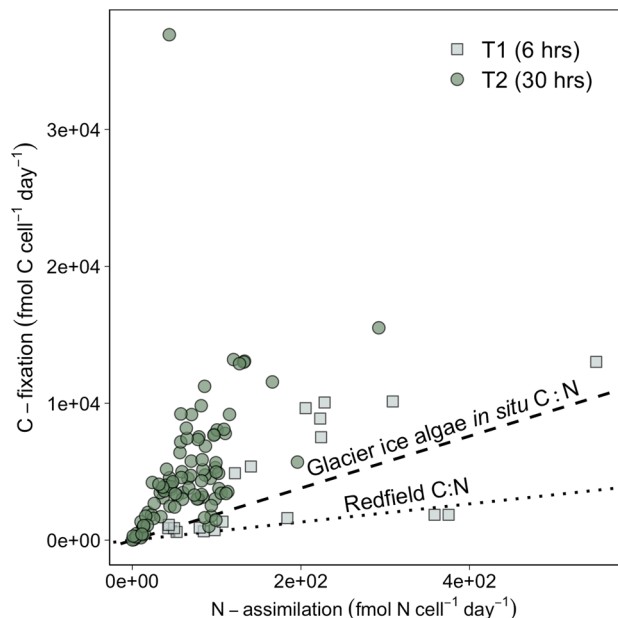

**Fig. 5 | Correlation between C-fixation and N-assimilation in active single glacier ice algal cells.** Squares represent T1 time points ($^{13}$C-DIC + $^{15}$NH$_4^+$ treatment; $n$ = 19), and circles represent T2 time points ($^{13}$C-DIC + $^{15}$NH$_4^+$ or $^{13}$C-DIC + $^{15}$NO$_3^-$ treatments; $n$ = 58). Dashed and dotted lines indicate the mean in situ C:N ratio of glacier ice algal cells (C:N = 19, see Fig. 2b) and the Redfield ratio (C:N = 6.6), respectively. Source data are provided as a Source Data file.

elemental composition of both the supraglacial community and individual glacier ice algal cells. Our findings suggest that the growth of glacier ice algae and the autotrophic microbial community is not limited by nutrient availability under the in situ conditions of our sampling site. Glacier ice algae efficiently assimilate available DIN sources are able to store excess P intracellularly, and exhibit variable and elevated C:N:P biomass ratios (mean of 509:26:1) compared to Redfield stoichiometries. These findings underscore the metabolic adaptations to low nutrient levels in situ and their potential to grow on new emerging bare ice surfaces using the allochthonous nutrients supplied by atmospheric deposition[38-41], surface ice melt[37] or N$_2$ fixation[38,68]. In the absence of other constraints on glacier ice algae abundance, such as infections by parasitic fungi[44,45] or trace metal limitation[20,29], algal blooms could form on newly exposed bare ice surfaces, leading to albedo reduction and enhanced surface ice melting, thereby constituting a potential positive feedback with climate warming. We reason, based on our findings, that the duration of bare ice exposure, and consequently the available growth period for glacier ice algae, maybe a primary factor in determining the extent of algal-driven ice darkening.

## Methods

### Study area and ice sampling

The supraglacial algal community was sampled near the SW tip of the Greenland Ice Sheet (61°05'8708"N, 46°50'9442"W), close to the PROMICE station QAS-M (61°05'54.7"N, 46°50'01.0"W), at an elevation of 680 m. On July 12, 2020, surface ice was collected by scraping off the top ~2 cm, which were placed into two 5 L Whirl-pack bags (Nasco, USA). The two bags were closed by wrapping the bag top over itself several times, likely sealing the bag from the exchange of gases with the atmosphere. The ice was allowed to melt under in situ light conditions at an ambient air temperature of ~4 °C. The ice took ~36 h to melt completely. The two bags of melted ice were combined into a single Whirl-pack bag, homogenized, subsampled and used for the incubations, as described below.

### Incubation experiment: C-fixation and N-uptake

We performed stable isotope incubation experiments to measure autotrophic C-fixation using $^{13}$C-DIC and N-assimilation from $^{15}$NO$_3^-$ and $^{15}$NH$_4^+$ in the supraglacial community. Additionally, we tested the combined effect of $^{15}$NH$_4^+$ and PO$_4^{3-}$, as well as the effect of PO$_4^{3-}$ alone (see Supplementary Fig. 3 for a graphical overview of the set-up). The melted surface ice, without any amendments, represents the T0 time point of our incubation experiment. The homogenized meltwater was then distributed into five 1 L blue cap Schott bottles, in which four different treatments and one control were prepared. C-fixation by algae was traced by adding 30 µmol L$^{-1}$ $^{13}$C-labelled bicarbonate ($^{13}$C-NaHCO$_3$, ≥98 $^{13}$C atom%; Sigma-Aldrich) to all treatments and control. N-assimilation was traced by adding $^{15}$N-labelled ammonium sulfate ($^{15}$N-(NH$_4$)$_2$SO$_4$) and $^{15}$N-labelled sodium nitrate ($^{15}$N-NaNO$_3$) to separate treatments (both ≥98 $^{15}$N atom%, Sigma-Aldrich) at ~10 µM final concentration. The effect of PO$_4^{3-}$ availability on C and N uptake was assessed by adding potassium di-hydrogen phosphate at a final concentration of 10 µM to separate $^{13}$C-DIC-only and $^{13}$C-DIC + $^{15}$N-NH$_4^+$ treatments. Once the tracers and nutrients were added, the liquid was gently homogenised and then further distributed from the 1 L Schott bottles into triplicated 250 mL serum bottles, closed with butyl rubber stoppers and aluminium crimps, leaving a 10 mL air head-space. These bottles were then incubated under in situ conditions on the ice surface for ~30 h, where they received a total amount of shortwave radiation of 346 W m$^{-2}$[69]. The 1 L Schott bottles with the remaining liquid for subsampling of the T1 timepoint were kept at ambient temperature and light, until sample processing finished.

Subsamples were taken at T0 from the melted glacier ice without nutrient or tracer addition, T1 (~6 h incubation time since nutrient or tracer addition, from the 1 L Schott bottles) and at T2 (~30 h incubation time, from the triplicate serum bottles). The following subsamples were taken: (1) for measurements of the atom% of the DIC pool after $^{13}$C-DIC addition (T1 and T2 time points), a liquid sample was collected with a syringe without headspace and bubble formation into 5.9 ml exetainers (Labco, Wales, UK), containing 100 µL saturated ZnCl$_2$ solution to stop the biological activity. The exetainers were stored in the dark at 4 °C until analysis; (2) For bulk C-fixation and N-assimilation measurements, the sample (145–193 mL) was filtered onto pre-combusted (450 °C) glass fibre filters (GF/F nominal pore size of 0.7 µm; Whatman, Maidstone, UK) and stored in plastic dishes at −80 °C until analysis. All bottles and laboratory equipment, such as filtration towers and forceps, were cleaned by soaking in 5% HCl overnight, followed by soaking and rinsing in Milli-Q. A filter rosette with one filter unit per treatment was used to avoid potential cross-contamination of isotopically labelled material; (3) For single-cell analyses by HR-SIMS (collected at T0 and T2, and for the control and $^{15}$N-NH$_4^+$ treatments at T1) and SEM-EDS (collected at T0), 5 mL sub-samples were collected and fixed with 2% EM-grade paraformaldehyde (PFA; EMS, USA) for 24 h at 4 °C. The fixed cells were then filtered onto 3 µm pore size gold–coated polycarbonate filters (25 mm diameter; GTTP, Merck Millipore, Eschborn, Germany), washed three times with ~10 mL of 0.2 µm filtered glacier stream water and stored at −20 °C; (4) Samples for dissolved inorganic and organic nutrient measurements (collected at T0, T1, T2) were collected by filtering 30 mL through 0.2 µm PES filters (25 mm diameter, Merck Millipore) with a polypropylene syringe into pre-washed 30 mL HDPE Nalgene bottles. To avoid any contamination for ultra-trace ion analysis, the bottles and caps were previously soaked in 5% HCl overnight and thereafter soaked in fresh Milli-Q water (Millipore, USA) for three days, with Milli-Q water replacement every day[70]. Once the nutrient samples were taken, the bottles were stored frozen at −20 °C until analysis; (5) For microscopy and cell counts (collected at T0), 2 mL of the sample liquid was preserved in duplicates in 2.5% EM-grade Glutaraldehyde (EMS, USA) and stored in the dark at 4 °C; (6) For amplicon sequencing (collected at T0), 500 mL of the melted ice surface sample was filtered onto a

sterile, 0.2 μm cellulose nitrate filter (Thermo Scientific Nalgene), which was preserved in a sterile cryotube, flash-frozen and transported to the home laboratory in a cryo-shipper. The filter was stored at −80 °C until nucleic acid extraction.

## Quantification of $^{13}C$-DIC, $^{15}N$-$NH_4^+$ and $^{15}N$-$NO_3^-$ atom% and atom%excess

The abundance of heavy stable isotope tracers, expressed in percentage ('atom%'), depends on the concentration of the added heavy isotope and its dilution with the naturally occurring isotopes of the same compound (DIC, $NH_4^+$, $NO_3^-$). The concentration of the heavy isotope corrected for the naturally occurring heavy isotope already present in the sample before tracer addition is termed 'atom% excess'. For determining the $^{13}C$-atom% of DIC, 2 ml subsamples from each $ZnCl_2$ fixed exetainer were injected into helium-flushed exetainers and acidified with phosphoric acid following Torres et al.[71] to convert all DIC to $CO_2$. Headspace subsamples were injected into GC-IRMS (isoprime precision, precision ± 0.1‰ for $^{13}C$-standards of 0–100 nM). The $^{13}C$-atom% of DIC in the ambient water was calculated from the measured concentrations of $^{13}C$-$CO_2$ and $^{12}C$-$CO_2$. Since the $^{13}C$-atom% of DIC between T1 and T2 decreased slightly (means of 4.4 to 3.6 atom% excess for all treatments), we used the mean value between T1 and T2 for each of the respective treatments for the T2 $^{13}C$-label incorporation calculations (3.9 $^{13}C$-atom% excess). For determining the $^{15}N$-atom% of $NH_4^+$ or $NO_3^-$, we used their added concentrations (10 μM with ≥98 $^{15}N$-atom%) and corrected it for the dilution with naturally occurring $NH_4^+$ or $NO_3^-$ at T0 (0.078 and 0.05 μM, respectively, with 0.36 $^{15}N$-atom% natural abundance), yielding 98 and 97 atom% excess for $^{15}N$-$NH_4^+$ and $^{15}N$-$NO_3$, respectively.

## Isotopic analyses of bulk particulate organic matter

The C and N contents and the isotopic composition of bulk particulate organic matter (at T0, T1 and T2) were determined from the particulate material collected on GF/F filters, which were dried at 60 °C, decalcified overnight under 37% HCl fumes in a desiccator and again dried at 60 °C. One-quarter of each filter was packed into tin capsules and analysed by an elemental analyser (Thermo Flash EA 1112) coupled to a continuous-flow Thermo Delta Plus XP isotope ratio mass spectrometer; Thermo Finnigan, Dreieich, Germany) (EA-IRMS) at the Max-Planck-Institute for Marine Microbiology (MPIMM), Germany. Caffeine was used as a standard for isotope ratio monitoring and C and N quantifications. The limit of detection (LOD) for isotopic enrichment was 1.078 $^{13}C$-atom% and 0.365 $^{15}N$-atom%.

## Single-cell elemental ratios and HR-SIMS analyses

Single-cell elemental ratios were obtained at the T0 timepoint using scanning electron microscopy (SEM, Quanta FEG 250, Thermo Fisher Scientific) coupled to energy-dispersive X-ray spectroscopy (EDS, Bruker Nano GmbH)[72–76], at the MPIMM. To avoid charging effects through the presence of large numbers of minerals, cells had to be transferred from the GTTP filters (3 μm) onto filters with a thicker gold coating prior to analysis (25 mm, 0.8 μm pore size, 40/20 nm coating; APC, Eschborn Germany). This was done by adding one drop of Milli-Q onto the filter surface with algal cells, placing the new filter piece with thicker gold coating onto a drop of water, freezing both filters together for 2 min and once frozen, removing the old filter by peeling it off. This procedure transferred substantial amounts of the original filter material onto the new filter surface without the need to scrape off any cells. Additionally, filters were gently rinsed with Milli-Q to remove some minerals/sediment grains. For morphological and autofluorescence-based identification of algal cells, the gold-coated filters were cut into sections (approx. 5 × 5 mm) and areas of interest were marked and imaged using a laser micro-dissection (LMD) microscope (6000 B, Leica) prior to SEM-EDS measurements. The EDS system is equipped with two QUANTAX XFlash 6/30 (Bruker Nano GmbH, Germany)

detectors. The detector area is 30 mm² and the detectors have an energy resolution at Mn K α line of <123 eV, allowing for the quantification of light elements. An NBS SRM 1155 ANSI 316 stainless steel standard was used to check the performance of the EDS system. 10 kV was used as a minimum accelerating voltage to analyse the sample for all major elements contained in the algal cells, also restricting the penetration depth to around 2 μm (demonstrated for cyanobacterial filaments in Schoffelen et al.[76]), so reducing any potential signal from the filter surface. The analysis of the elemental content of algal cells was performed using the standardless P/B-ZAF method (Quantax 400 software, version 1.9; Bruker), suitable for samples with topography and allowing for measurements of light to heavy elements. Further details on the data processing can be found in Khachikyan et al.[73]. Cells which were too thin for a robust signal were excluded from data processing by manually inspecting the obtained spectra. Single-cell relative C:P, C:N, and N:P atomic ratios were determined from the measurements in atom%, while the data in mass% was used to calculate the absolute elemental content of algal cells (see next section).

The T0 sample, one replicate per treatment of the T2 timepoint, and additionally the T1 of the control and the $^{15}N$-$NH_4^+$ amended treatments (due to rapid $NH_4^+$ cycling), were used for HR-SIMS analysis. The pre-imaged filter pieces from SEM-EDS analysis and additional filter pieces were mounted on a glass slide and coated with a 5 nm layer of gold prior to HR-SIMS analyses. Single-cell $^{15}N$ and $^{13}C$ assimilation rates of algal cells were determined by HR-SIMS (IMS 1280, CAMECA, Gennevilliers, France) at the Natural History Museum in Stockholm, Sweden. Areas of interest were pre-sputtered with a primary Cs⁺ ion beam of 3 nA for 240 s over an area of 80 × 80 μm and then analysed with a 100 pA beam over 70 × 70 μm at a spot size of 1 μm for 60 cycles. The HR-SIMS images (256 × 256 pixel) were recorded for $^{12}C^{15}N^-$, $^{13}C^{14}N^-$ and $^{12}C^{14}N^-$ ions with a peak-switching routine at a mass resolving power of 12,000 (M/ΔM) using a low-noise ion-counting electron multiplier. The detection limit was <0.01 counts per second (cps). For integration times of 60 s ($^{12}C^{14}N^-$), 300 s ($^{12}C^{15}N^-$) and 120 s ($^{13}C^{14}N^-$) over 60 cycles, a run was expected to have total background count lower than 0.6, 3 and 1.2, respectively, not requiring any baseline correction. For the 256 × 256 pixel resolution, this approximates to background levels of $1e^{-5}$, $5e^{-5}$ and $2e^{-5}$ cps pixel⁻¹, respectively. Images were processed using the CAMECA WinImage2 software. Secondary ion images were drift-corrected and accumulated for each measurement and the detector dead time, electronically gated at 44 ns, was processed on each pixel. Regions of interest (ROIs) were manually drawn around the algal cells. The $^{13}C/(^{13}C + ^{12}C)$ and $^{15}N/(^{15}N + ^{14}N)$ ratios were subsequently calculated as means for each ROI. Unlabelled (natural abundance) glacier ice algae cells from non-incubated samples were also measured ($n = 29$) and mean isotope fractions (0.0037 ± 0.00006 and 0.0111 ± 0.00016 for $^{15}N$ and $^{13}C$, respectively) were subtracted from the labelled samples to obtain 'excess' isotope fractions of the biomass. *A. alaskanum* and *A. nordenskioeldii* are grouped together as glacier ice algae within this study, due to their taxonomically close relationship and partial size overlap[77], which challenged an unambiguous species identification from microscopic images obtained for the filtered cells. We acknowledge that the fixation of the algal cells with PFA after incubations for HR-SIMS analysis may result in a decrease of $^{13}C$-enrichment of ca. 4–8%, and, to a lesser extent $^{15}N$-enrichment[78–80]. However, this effect is likely considerably smaller than the differences observed in the C- and N-based growth rates between single-cell (fixed with PFA) and bulk (preserved by freezing) measurements in our study. We therefore chose not to apply any corrections to the measured enrichment values of the single-cell analyses. Cells were considered as enriched/active if their mean $^{13}C/(^{13}C + ^{12}C)$ enrichment exceeded the mean observed natural abundance value + 3x the standard deviation of unlabelled control cells[81] (1.15 $^{13}C$-atom% for glacier ice algae).

## Cellular biovolume, dry weight, and absolute elemental content of glacier ice algal cells

Cell dimensions were obtained from HR-SIMS images using ImageJ. Biovolumes were subsequently calculated by assuming cylindrical shapes for glacier ice algae after Hillebrand et al.[82]. Cellular dry weights (pg cell$^{-1}$) were calculated by multiplying the algal biovolumes (mean of $1414 \pm 873$ μm$^{-3}$ for all imaged algal cells, $n = 244$) by the glacier ice algal-specific buoyant density of 1160 kg m$^{-3}$ [83] and a mean dry fraction of 0.28 (obtained from *C. vulgaris*[84]). Absolute elemental contents of glacier ice algal cells (pg element cell$^{-1}$) were determined by multiplying the median mass fraction of C, N or P in glacier ice algal cells (0.72, 0.04 and 0.04 C, N and P, respectively, derived from SEM-EDS, Supplementary Table 5) by the cellular dry weights (pg cell$^{-1}$)[73].

## C- and N-assimilation rates determined by EA-IRMS and HR-SIMS

Bulk C-assimilation rates were calculated using the following equation[85]:

$$C\ assimilation\ rate\left(\mu mol\ C\ L^{-1}\ day^{-1}\right) = \frac{^{13}C\ atom\%\ excess_{POC}}{^{13}C\ atom\%\ excess_{DIC}} \times \frac{POC}{\Delta t} \quad (1)$$

where $^{13}$C-atom% excess$_{POC}$ represents the $^{13}$C-atom% of incubated POC minus its natural abundance atom%, *POC* refers to the biomass concentration (μmol C L$^{-1}$), $^{13}$C-atom% excess$_{DIC}$ represents the $^{13}$C-atom% in DIC minus its natural abundance atom% and $\Delta t$ represents the incubation period (in days, T0-T1 or T0-T2). We assume that the $^{13}$C-assimilation rates correspond to net photosynthesis, as any $^{13}$C fixed during the incubation (1.1 days including ~6 h of twilight) may have partially been respired again, which would not be measured by HR-SIMS.

Bulk N-assimilation rates from NH$_4^+$ or NO$_3^-$ were calculated analogously from the $^{15}$N-atom% of incubated PON minus its natural abundance atom%, the corresponding PON concentration of the sample (μmol N L$^{-1}$), the $^{15}$N-atom% of NH$_4^+$ or NO$_3^-$ present in the incubation water minus their natural abundance atom%, and the incubation period, as described above.

Single-cell specific C-fixation rates were calculated according to the following equation[85]:

$$C\ fixation\ rate\left(pmol\ C\ cell^{-1}\ day^{-1}\right) = \frac{^{13}C\ atom\%\ excess_{cell}}{^{13}C\ atom\%\ excess_{DIC}} \times \frac{C_{cell}}{\Delta t} \quad (2)$$

where $^{13}$C-atom% excess$_{cell}$ represents the $^{13}$C-atom% of single algal cells minus their natural abundance atom%, $C_{cell}$ represents the mean C content of single algal cells (pmol C cell$^{-1}$, calculated as described above), $^{13}$C-atom% excess$_{DIC}$ represents the $^{13}$C-atom% in DIC minus its natural abundance atom% and $\Delta t$ represents the incubation period (in days, T0-T1 or T0-T2).

Single-cell specific N-assimilation rates from NH$_4^+$ or NO$_3^-$ were calculated analogously from the $^{15}$N-atom% of single algal cells minus the natural abundance atom%, the $^{15}$N-atom% of NH$_4^+$ or NO$_3^-$ present in the incubation water minus the natural abundance atom%, the corresponding mean N content of single algal cells (pmol N cell$^{-1}$, calculated as described above), and the incubation period, as described above.

## C-and N-based growth rates

Growth rates based on $^{13}$C-DIC, $^{15}$NH$_4$ or $^{15}$NO$_3$ isotope uptake were calculated for the bulk community (EA-IRMS measurements) or single algal cells (HR-SIMS measurements). C-based growth rates (day$^{-1}$) were

calculated following Martínez-Pérez et al. [74], based on Montoya et al. [85]:

$$
\begin{aligned}
&C - based\ growth\ rate\left(day^{-1}\right) = \\
&\log_2\left[\frac{^{13}C\ atom\%\ excess_{DIC}}{\left(^{13}C\ atom\%\ excess_{DIC} - {}^{13}C\ atom\%\ excess_{POC}\right)}\right] \times \frac{1}{\Delta t}
\end{aligned} \quad (3)
$$

where $^{13}$C-atom% excess$_{DIC}$ represents the $^{13}$C-atom% in DIC minus its natural abundance atom%, $^{13}$C-atom% excess$_{POC}$ the $^{13}$C-atom% of incubated POC (of either bulk or single-cell biomass) minus its natural abundance atom%, and $\Delta t$ representing the incubation period (in days, T0-T1 or T0-T2).

N-based growth rates were calculated analogously from the $^{15}$N-atom% excess of either NH$_4^+$ or NO$_3^-$ in the incubation water, the $^{15}$N-atom% excess of PON of either bulk or single-cell biomass, and the incubation period, as described previously. The C or N-based growth rates assume exponential growth[74] and that all newly incorporated $^{13}$C or $^{15}$N are due to biomass increase[86], e.g. a growth rate of 1 day$^{-1}$ means that cells double their C or N content once per day and, thus, divide once. The obtained growth rate estimates are independent of the biomass[85]. A fraction of assimilated $^{13}$C or $^{15}$N may be allocated to C- or N-storage, recycling or replacing of cell components without net per cell growth. However, as this fraction is unknown, we do not consider it in our calculations. See Polerecky et al. [65] and Halbach[66] for more details on assumptions for isotope uptake calculations. Population doubling times were calculated as 1/growth rate.

## Glacier ice algae contribution to bulk C- and N-uptake

Similar to previous studies[34,74], we estimated the relative contribution by active glacier ice algae to the total bulk C and N uptake (originating from $^{13}$C-DIC or $^{15}$NH$_4^+$) for the different timepoints:

$$Rel.\ contribution\ (\%) = \frac{(assimilation_{cell} \times N_{cell})}{assimilation_{bulk}} \times 100 \quad (4)$$

where assimilation$_{cell}$ is the mean assimilation rate of active glacier ice algae of the respective substrate (pmol element cell$^{-1}$ day$^{-1}$), N$_{cell}$ is the mean abundance of the active glacier ice algae (cells L$^{-1}$) and assimilation$_{bulk}$ represents the assimilation rates of the bulk community of the corresponding time point (μmol element L$^{-1}$ day$^{-1}$). The active glacier ice algal cell numbers are derived from algal counts at T0, corrected for the active population fraction based on SIMS measurements of C fixation (90% active cells). Biomass distribution between incubation bottles was variable due to rapid sinking of particulate material, thus, the large uncertainty associated with the parameter of assimilation$_{bulk}$ contributes to the uncertainty of relative contribution by glacier ice algae. To account for varying biomass between bottles and the potential varying algal abundance, we also performed the calculations using the algal abundance corrected by the fractional change in POC concentrations between T0 and T1, as well as T0 and T2. This revealed a consistently low contribution (7–12% for C from DIC and 3–4% for N from NH$_4^+$; Supplementary Tables 2 and 3). Uncertainties in the contribution of the glacier ice algal community assimilation to total assimilation derive from the combined uncertainties of each variable, following the laws of error propagation (Supplementary Note 2).

## Dissolved nutrient analysis

Dissolved NO$_3^-$, NO$_2^-$, NH$_4^+$, and PO$_4^{3-}$ concentrations from T0, T1 and T2 were analysed on a Metrohm Ion chromatography system (883 Basic IC Plus and 919 Autosampler Plus) at Uppsala University, Sweden. The IC was equipped with a peristaltic pump to enable full loop injections (400 μl) to decrease the LOD and limit of quantifications (LOQ)[70]. Sample tubes were stored with a lid in the autosampler to avoid contamination with N from air. LOD's and LOQ's were

determined as 3× and 10× the standard deviation (STDEC) of the lowest nutrient concentrations from standards, according to the EPA procedure for method detection limit[87]. LOD's were 0.011, 0.008, 0.005 and 0.004 μM and LOQ's were 0.022, 0.027, 0.018 and 0.007 μM, for $NO_2^-$, $NO_3^-$, $PO_4^{3-}$, and $NH_4^+$, respectively. The corresponding mean precisions were ±3, ±8, ±5 and ±3% and accuracy −8, −12, −4 and −1%, for $NO_2^-$, $NO_3^-$, $PO_4^{3-}$, and $NH_4^+$, respectively, as determined from a comparison of QC standards with 0.043, 0.026, 0.015, and 0.015 μM levels.

Total dissolved nitrogen (TDN) was analysed on a Shimadzu TNM (Tokyo, Japan). DON was calculated as DON = TDN−DIN, where DIN is ($NO_3^-$ + $NH_4^+$ + $NO_2^-$). Total dissolved phosphorus (TDP) was analysed by the molybdenum blue method after digestion with potassium persulfate and autoclaving at 121 °C for 60 min. DOP was then calculated as DOP = TDP − $PO_4^{3-}$. There was insufficient liquid left from the first collected sample (T0) and the reported concentration was measured from a sampling location close to the experimental site, but three days later. The LODs were 0.83 and 0.03 μM and LOQs 0.03 and 0.07 μM for TDN and TDP, respectively. The accuracy for TDN was 12% and precision 5% with a standard of 0.03 μM N.

The T0/in situ samples for analysis of $Na^{2+}$, $Mg^{2+}$, $K^+$ and $Ca^{2+}$ were acidified using Aristar $HNO_3$. The analyses of major, minor and trace element analyses were carried out with an inductively-coupled plasma mass spectrometer (ICP-MS; Thermo Fisher iCAPQc). The precision of the analyses was between 1–5% and LOD's for $Na^{2+}$, $Mg^{2+}$, $K^+$ and $Ca^{2+}$ were 0.2, 0.03, 0.65 and 0.46 μg $L^{-1}$, respectively. The ICP-MS analyses were conducted by Stephen Reid at the University of Leeds, UK.

## Community composition and algal abundance

The algal abundance and community composition at T0 were microscopically characterised from the glutaraldehyde preserved samples and algal cells counted on a B/W FlowCAM™ II (Fluid Imaging Technologies, Maine, USA) using a 100 μm × 2 mm flow cell, a 10× objective and the automated-imaging mode. A minimum of 760 total algal cells per sample were counted. Algal cells were subsequently taxonomically identified, using the VisualSpreadSheet (VISP). Additional pictures of the supraglacial community were taken from unfixed, fresh sample material using a Nikon Eclipse Ti microscope. Images of the fresh unfixed and fixed algal cells were screened for signs of fungal infections (Supplementary Fig. 2).

Amplicon sequencing was performed to determine the microbial composition of the sample prior to incubation. DNA extraction was performed using the DNeasy PowerSoil Pro Kit (Qiagen) according to the manufacturer's protocol. Thereafter, DNA concentration was measured on a Qubit 3.0 (Invitrogen) with the broad-range dsDNA kit (Invitrogen). Amplification was performed for the bacterial 16S rRNA gene using Bakt_341F (5′- CCTACGGGNGGCWGCAG-3′) and Bakt_805R (5′- GACTACHVGGGTATCTAATCC-3′)[88] and for the 18S rRNA gene using 528 F (5′- GCGGTAATTCCAGCTCCAA-3′) and 706 R (5′-AATC-CRAGAATTTCACCTCT-3′)[89]. The amplicon library was built in a two-step PCR. Each reaction of the first PCRs contained 12.5 μL of 2x PCRBIO Ultra Mix (PCR Biosystems), 0.5 μL of forward and reverse primer from a 10 μM stock, 0.5 μL of bovine serum albumin (BSA) to a final concentration of 0.025 mg $mL^{-1}$, 0.6 μL of sterile water and 5 μL of template DNA. For the first PCR, conditions were as follows: at 95 °C for 2 min, followed by 38 cycles of 95 °C for 15 s, 55 °C for 15 s, 72 °C for 40 s, with a final extension performed at 72 °C for 4 min. An electrophoresis 1% agarose gel was run with PCR products before proceeding. Samples were subsequently indexed in a second PCR. In the second PCR, 5 μl of product from the first PCR was used as template to add indexes and sequencing adaptors in a reaction consisting of 12.5 μl of 2× PCRBIO Ultra Mix (PCR Biosystems), 2 μl of each index primer (P5/P7), and DNase free water to a final volume of 28 μl. For the second PCR, conditions were as follows: pre-incubation at 98 °C for 1 min, followed by 13 cycles of 98 °C for 10 s, 55 °C for 20 s, and 72 °C for 40 s, and ending with a final step at 72 °C for 5 min.

The final PCR products were purified with 15 μl HighPrep PCR magnetic beads (MagBio Genomics Inc. Gaithersburg, Maryland, US) according to the manufacturer's instructions and eluted in 27 μl TE buffer. Aliquots of the PCR products were run on a 1.5% agarose gel and checked under UV light. Concentrations of the amplified and purified DNA samples were measured on a Qubit 2.0 fluorometer (Invitrogen, Eugene, Oregon, US). The samples were then equimolarly pooled, and this final library was sequenced on an Illumina MiSeq using the V2 kit (Illumina Inc. SanDiego, California, US) resulting in 2 × 250 bp reads.

## Data analysis

Statistical analysis and plotting was undertaken in R (version 4.4.2)[90]. The non-parametric Kruskal-Wallis $t$-test was used to explore the similarity of data for individual treatments generated by HR-SIMS and EA-IRMS, followed by a post-hoc test of multiple comparisons using the Fisher's least significant difference criterium and Holm's p-adjustment method. Data were considered significantly different at $p < 0.05$. Inactive cells (i.e. those with no significant $^{13}C$-DIC incorporation) were excluded from statistical tests involving cell activity. Results are presented as mean ± standard deviation. The 16S and 18S rRNA gene amplicons were pre-processed and analysed using the DADA2 R package[91] for ASVs. Taxonomic assignment was made using the SILVA (V148) rRNA gene database[92]. Detailed documentation of the pipelines, including parameter setups, is available in Trivedi et al. [93]. Results were visualized using the phyloseq v1.36.0[94] and ggplot2 v3.3.5 R packages[95]. Classes and phyla with <1% mean relative abundance were grouped as Others for 16S and 18S rRNA gene data representation, respectively.

## Reporting summary

Further information on research design is available in the Nature Portfolio Reporting Summary linked to this article.

# Data availability

Data analysed in this study are included in the Supplementary Information and source data file. The amplicon sequencing data are available under BioProject ID PRJNA1209368 for the 16S data, and under BioProject ID PRJNA1209915 for the 18S data. Source data are provided with this paper.

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

## Acknowledgements

The presented work is part of the project DeepPurple which has received funding from the European Research Council (ERC) under the European

Union's Horizon 2020 research and innovation programme (Grant Agreement No. 856416). Alexandre Anesio and Martin Hansen received support from the Aarhus University Research Foundation (grant numbers AUFF-T-2017-FLS-7-4 and AUFF-2018), Laura Halbach, Katharina Kitzinger, Sten Littmann and Marcel M. M. Kuypers from the Max Planck Society. Liane G Benning and Rey Mourot were also supported through funding from The Helmholtz Recruiting Initiative (award no. I-044-16-01). James A Bradley was supported by the Agence Nationale de la Recherche (ANR23-CPJ1-0172-01), the Alexander von Humboldt Foundation, and the European Research Council (ERC) under the European Union's Horizon Europe Research and Innovation programme (Grant agreement No. 101115755). We would like to thank Swantje Lilienthal for their help during the SEM-EDS imaging and Gabriele Klockgether, Wiebke Mohr and Gaute Lavik for fruitful discussions. We would like to thank Marie Bolander Jensen for organising the analysis of the amplicon sequencing and Christoffer Bergvall for analysing the nutrient samples. We acknowledge NordSIMS-Vegacenter for the provision of facilities and experimental support, and we thank Kerstin Lindén and Heejin Jeon for their assistance. NordSIMS-Vegacenter is funded by the Swedish Research Council as a national research infrastructure (Dnr. 2021-00276) and is further supported by the Swedish Museum of Natural History and the University of Iceland. Data from the Programme for Monitoring of the Greenland Ice Sheet (PROMICE) were provided by the Geological Survey of Denmark and Greenland (GEUS) at http://www.promice.dk.

## Author contributions

The study was designed by LH, KK, AMA, MH, JAB and LGB. LH and AMA conducted the experiments. LH drafted the manuscript, collected the samples, and analysed the data. KK helped with study design and sampling protocols, conducted the IRMS analysis and helped with data interpretation. AMA helped in study design, sample collection and data interpretation. MJW conducted the HR-SIMS analysis and helped during data analysis and interpretations. SL conducted the SEM-EDS analysis, and its data analysis and helped with their interpretations. MH helped during the study design and data interpretations. MMMK supported the study design and provided funding for SEM-EDS and IRMS analyses. LGB helped with study design and data interpretations. RM helped with sample collection, in-field processing as well as DNA extraction and analysis of the sequencing data. MO and JAB helped in the analysis of stable isotope data and its interpretations. MT helped with data interpretation. LEJ supervised DNA library building, sequenced the amplicon libraries and helped during their analysis. MT, LGB and AMA obtained funding for the Deep Purple ERC Synergy project. All co-authors contributed to the drafting of this manuscript.

## Funding

## Competing interests

The authors declare no competing interests.
