## [Transparent Peer Review file · Nature Communications]

Single-cell imaging reveals efficient nutrient uptake and growth of microalgae darkening the Greenland Ice Sheet

Corresponding Author: Dr Laura Halbach

Version 0:

Reviewer comments:

Reviewer #1

(Remarks to the Author)

Halbach et al. present a rigorous investigation of macronutrient uptake and bulk growth rate metrics, combined with exceptional individual cell imaging to understand if macronutrient amendments stimulate growth and biochemical reconfigurations within. This is an incredibly detailed study and warrants publication after consideration of comments and additional discussion. The paper is well-written and experimental setup is thoroughly complete. Since their recognition as important regulators of ice sheet melt balance (Halbach et al., 2022), it is fantastic to see this work attempt to test various hypotheses about the biogeochemical controls. Although, I believe that this work represents just the beginning. It is, however, a great contribution to the field.

Overall, I do believe this is a unique study with important implications for the success and fitness of a foundational algal species. I think the results in the study support their conclusion that the cells are efficient at optimizing their cellular demand for N, while demonstrating luxury uptake of P and storage as granules within the cell. I do believe, however, that given these algae are probably limited by the transport of P-rich minerals to the ice, this would also include micronutrients, which were not investigated. I hope that the authors consider my comments as a possible future direction for understanding the bottom-up control of inorganic nutrients.

The non-Redfieldian ratios highlighted in their study could use some more discussion about what kind of biomolecules are contributing to elevated C:N:P. For example, the production of pigments or polymeric molecules would tend towards high C:N:P. What is known about their lipid content to handle the cold temperatures? I felt discussion of the physiological traits to be quite limited.

Given that this is an excellent study, and their results are irrefutable in light of the amount of data presented, I raise the question, what are these algae limited by? It is hard to argue that light might limit their growth on a highly reflective surface with 18 hours of sunlight per day. In fact, this intensity of irradiation could cause detrimental effects through inefficient electron transport (production of ROS) or excessive heat (although this is compensated by the cold temperatures, see Strzepek et al., 2019). Inefficient light harvesting could lead to higher rates of mortality, and necessarily drive quotas for electron transporters or superoxide dismutase to be higher, certainly among the microbial community. Meta-transcriptomics would be very useful to tease apart these biochemical responses. I wonder if there exists an ROS process that effectively caps the ability for the organisms to grow given these nutrient amendments. I think it is possible that a mechanism to retain essential micronutrients on the ice surface would be prerequisite for efficient N and P uptake/storage. There is no large contribution of micronutrients from meltwaters (glacial ice, besides basal ice in contact with bedrock, has very low concentrations of soluble iron, see Hopwood et al., 2017), that micronutrients such as iron could be a limiting factor. It seems like iron could be diverted within the cell to uptake nitrate. Without sufficient iron for nitrate reductase, nitrate assimilation could be inhibited. Further, such efficient regeneration of nutrients in oligotrophic regions require a mechanism to retain iron to support high rates of heterotrophy and turnover of DON and DOP (although alkaline phosphatases are typically Zn- and Mg- dependent). This points towards an emergent hypothesis that turnover of organic pools of N and P are extremely important in this system, and so the growth of glacier ice algae is dependent on close symbioses with heterotrophs – as alluded to in the text. I am currently unsure how diffusion might affect these transient communities living in meltwater pools or ice surfaces. In other aquatic environments, it is presumably more turbulent, and so the flux of nutrients to the cell surface is greater than on the ice surface.

While such a study on micronutrients is outside of the scope of this paper, I do think it is worth a short discussion as to the

role of trace-metals in supporting communities optimized for low macronutrient conditions. In the very least, it does appear that the glacier ice algae are able to divert intracellular metals toward N-assimilation instead of C-fixation. Again, meta-transcriptomics would be immensely helpful to investigate responses to nutrient amendments.

Specific in-line comments:

[99-100] remove "the yields insights into"

[172] I was hoping that cell-specific pigments would also be measured as it could drive the variability seen between cells, although I'm not sure how that would change over 30 hours.

[Fig. 4] Are there no T1 single cell C or N based growth rates for nitrate and phosphate? Alternatively, should the ammonium be calculated as a rate from T0 given that results for T1 are displayed, or is that for consistency to the other nutrient amendments?

[241] So it does appear they are approaching Redfield ratio as the allocation of C and N are reconfigured in the cell. This is shown in Fig. 5, where the data, still with lots of scatter, shows that at high nutrient additions, they opportunistically take up N into the cell, but that does not translate to higher C fixation. At low N availability, C-fixation drops to where many cells are close to Redfield. However, most cells lie above this line, so I agree with the authors interpretation.

[311] The reference requires another bracket "[47]"

[346] DOC retained on filters might require a reference, if that is known.

References:

Strzepek, R. F., Boyd, P. W., and Sunda, W. G., 2019, Photosynthetic adaptation to low iron, light, and temperature in Southern Ocean phytoplankton: Proceedings of the National Academy of Sciences, v. 116, p. 4388–4393.

Hopwood, M. J., Cantoni, C., Clarke, J. S., Cozzi, S., and Achterberg, E. P., 2017, The heterogeneous nature of Fe delivery from melting icebergs: Geochemical Perspectives Letters, v. 3, p. 200–209.

Reviewer #2

(Remarks to the Author)

The manuscript presents single-cell data on elemental composition and C and N assimilation of glacial ice algae, providing important new insights into their ecophysiology as well as the ecological and biogeochemical functioning of this system. The results are presented clearly, experiments seem mostly carefully conducted and discussion and conclusions adequate, however I have one concern regarding the experimental design that requires clarification and some questions/suggestions regarding interpretation (see below)

Specifically, based on the text in l. 420-421, and l. 426, do I understand it correctly that 1L of sample was spiked in a Schott bottle, then 750 ml were transferred to serum bottles for 30h incubations, while ca 250 ml remained in the Schott bottle for the 6h incubation? If so, it seems the conditions in 6h and 30h incubations were quite different (especially given the large headspace in the Schott bottle which, for instance, may have affected gas exchange, incl. DIC and ¹³C labelling). Couldn't this difference in incubation conditions explain the difference in C and N assimilation rates between t1 and t2?

The authors conclude from the nutrient addition experiments that the algae are not nutrient limited (e.g. l. 386) and further that they can readily colonize new ice surfaces (ll. 390-394) – I guess nutrients are only part of the picture, doesn't this also depend on ecological interactions (i.e. how growth rates compare to those of potential competitors)? Also, at least in theory the lack of a response to the nutrient additions could indicate that some other (micro)nutrient (other than those tested) is limiting, can this be briefly discussed?

Curiously, the study shows lower growth rates in nutrient-amended bottles than in controls. As an explanation, in l. 301 it is suggested that these algal populations might be sensitive to high nutrient loads – what would be the mechanism here, could this be explained in a little more detail?

Other minor comments:

Table 2 shows mean values for two time points but SD only for one of them. I suggest showing both time points with their respective SD.

fig. S2b and fig. 1c are the same – I believe one could be omitted?

Also, I think it would be helpful to indicate any chytrids in the microscope images by arrows. To non-specialists it is not clear whether the spherical 'cells' are algal cysts or chytrids

l. 370 discusses the role of variable activity for adapting to environmental gradients over time - what about spatial (micro)environmental gradients in the ice, are those relevant?

Table 1 – state the units

l. 99 typo – delete 'highlights the'

l. 339-341 revise phrasing, 'despite' does not seem to make sense

l. 477-478 typo - delete 'again'

l. 776 journal name is missing

l. 930, 931 please correct spelling of N₂ fixation

Supplement l. 24 'assumingly' - presumably?

Version 1:

Reviewer comments:

Reviewer #2

(Remarks to the Author)

Thank you for carefully considering my comments on the previous version of the manuscript. My questions and concerns have been adequately addressed. I believe the revisions have improved the manuscript and look forward to seeing it published.

Response to reviewer comments by Halbach et al.

Response to reviewers' comments on '**Single-cell imaging reveals efficient nutrient uptake and growth of microalgae darkening the Greenland Ice Sheet**'.

We would like to thank the reviewers for their helpful and constructive comments and suggestions. We have addressed all issues that were raised, which further improved the manuscript. Please find below our responses to the specific points. In our resubmission, we provide a clean, updated manuscript file, as well as one where all changes are indicated. The line numbers mentioned here refer to the "clean version" of the manuscript file without markup.

REVIEWER COMMENTS

Reviewer #1 (Remarks to the Author):

Halbach et al. present a rigorous investigation of macronutrient uptake and bulk growth rate metrics, combined with exceptional individual cell imaging to understand if macronutrient amendments stimulate growth and biochemical reconfigurations within. This is an incredibly detailed study and warrants publication after consideration of comments and additional discussion. The paper is well-written and experimental setup is thoroughly complete. Since their recognition as important regulators of ice sheet melt balance (Halbach et al., 2022), it is fantastic to see this work attempt to test various hypotheses about the biogeochemical controls. Although, I believe that this work represents just the beginning. It is, however, a great contribution to the field.

Overall, I do believe this is a unique study with important implications for the success and fitness of a foundational algal species. I think the results in the study support their conclusion that the cells are efficient at optimizing their cellular demand for N, while demonstrating luxury uptake of P and storage as granules within the cell. I do believe, however, that given these algae are probably limited by the transport of P-rich minerals to the ice, this would also include micronutrients, which were not investigated. I hope that the authors consider my comments as a possible future direction for understanding the bottom-up control of inorganic nutrients.

We thank the reviewer for their encouraging comments and interesting thoughts! We agree that future studies are needed to shed light on the actual controls of ice algae growth, and have included a discussion of possible future research avenues in the revised manuscript (please see detailed answers below), while trying to stay within the word limit and keeping the manuscript still focused on macronutrient limitation.

The non-Redfieldian ratios highlighted in their study could use some more discussion about what kind of biomolecules are contributing to elevated C:N:P. For example, the production of pigments or polymeric molecules would tend towards high C:N:P.

What is known about their lipid content to handle the cold temperatures? I felt discussion of the physiological traits to be quite limited.

Investigation of the specific physiological adaptations and which specific compounds cause the elevated C:N:P content lies outside the scope of this paper. However, we agree with the reviewer that biomolecule accumulation may contribute towards the elevated C:N:P content in glacier ice algae, and we have edited the manuscript to explicitly discuss this (line: 326), as it provides an important avenue for future investigations. We agree that the C-rich phenolic pigments likely contribute significantly to the elevated C content (lines: 283-290). Peters et al. (2024) studied the lipids in glacier ice algal dominated algal communities, and that study shows that lipids dominate the endometabolites. We added the reference to the

text (line 287). It is likely that lipids play an important role in the adaptation by glacier ice algae to the cold temperatures of their habitat.

Given that this is an excellent study, and their results are irrefutable in light of the amount of data presented, I raise the question, what are these algae limited by?

It is indeed an intriguing question of what the ultimate limiting factors for ice-algae growth are, and we have now incorporated a more extended discussion of potential controls in the revised manuscript (discussion: lines 247-253). Please additionally see our detailed answers to potential controls below.

It is hard to argue that light might limit their growth on a highly reflective surface with 18 hours of sunlight per day. In fact, this intensity of irradiation could cause detrimental effects through inefficient electron transport (production of ROS) or excessive heat (although this is compensated by the cold temperatures, see Strzepek et al., 2019). Inefficient light harvesting could lead to higher rates of mortality, and necessarily drive quotas for electron transporters or superoxide dismutase to be higher, certainly among the microbial community. Meta-transcriptomics would be very useful to tease apart these biochemical responses. I wonder if there exists an ROS process that effectively caps the ability for the organisms to grow given these nutrient amendments.

We agree that the ice-algal cells may experience stress from excess irradiation, and that this could contribute to their mortality (we estimate that ca. 10% of the ice-algal cells are not active), in addition to the observed fungal infections. Future metatranscriptomics analyses looking specifically into regulation of ROS defense mechanisms under varying light conditions, as well as direct measurements of ROS concentrations in the melt water, could indeed yield further insights into this.

I think it is possible that a mechanism to retain essential micronutrients on the ice surface would be a prerequisite for efficient N and P uptake/storage. There is no large contribution of micronutrients from meltwaters (glacial ice, besides basal ice in contact with bedrock, has very low concentrations of soluble iron, see Hopwood et al., 2017), that micronutrients such as iron could be a limiting factor. It seems like iron could be diverted within the cell to uptake nitrate. Without sufficient iron for nitrate reductase, nitrate assimilation could be inhibited. Further, such efficient regeneration of nutrients in oligotrophic regions require a mechanism to retain iron to support high rates of heterotrophy and turnover of DON and DOP (although alkaline phosphatases are typically Zn- and Mg- dependent). This points towards an emergent hypothesis that turnover of organic pools of N and P are extremely important in this system, and so the growth of glacier ice algae is dependent on close symbioses with heterotrophs – as alluded to in the text.

Both reviewers mentioned the possibility of micronutrients to be acting as potential controls of glacial algae growth. Although this was not tested in our experimental setup, we fully agree that limitation by micronutrients is very important to be considered. As the reviewer points out, micronutrients such as iron and other metals could be a key factor limiting algal growth, and the mechanisms for retaining and efficiently using these within the ice surface community should be addressed in future studies. These could include studying micronutrient availability on a microscale in relation to biofilm formation and DON/DOP turnover, combined with metatranscriptomic analyses of the community under different micro- and macronutrient-loading regimes. We have, therefore, added a section in the discussion specifically about micronutrients (discussion: lines 247-253).

I am currently unsure how diffusion might affect these transient communities living in meltwater pools or ice surfaces. In other aquatic environments, it is presumably more turbulent, and so the flux of nutrients to the cell surface is greater than on the ice surface.

This is an interesting point. How advection and turbulence are impacting the nutrient fluxes on the ice surface deserves further studies. Certainly, continuous melting of the ice during the daytime could supply nutrients. The biofilm around the algal communities and other microorganisms present also needs to be considered when trying to understand the nutrient fluxes on the microscale. Note, however, that we have refrained from speculating too much in the paper.

While such a study on micronutrients is outside of the scope of this paper, I do think it is worth a short discussion as to the role of trace-metals in supporting communities optimized for low macronutrient conditions. In the very least, it does appear that the glacier ice algae are able to divert intracellular metals toward N-assimilation instead of C-fixation. Again, meta-transcriptomics would be immensely helpful to investigate responses to nutrient amendments.

We appreciate the thoughts of the reviewer on potential factors controlling ice-algae growth, and, as suggested, we have added a new discussion point of the potential relevance of trace metal cycling in glacier ice algae habitats to the revised manuscript (discussion: lines 247-253, and conclusion 353).

Specific in-line comments:
[99-100] remove “the yields insights into”

Agreed and revised accordingly.

[172] I was hoping that cell-specific pigments would also be measured as it could drive the variability seen between cells, although I'm not sure how that would change over 30 hours.

We agree that the observed inter-cellular variability in pigmentation (Figure 1c and Figure S2) is highly interesting and could explain the observed differences in single-cell activities. Unfortunately, using the methods available to us (and the samples that were taken during the study), we were not able to resolve pigment type or content of individual cells.

[Fig. 4] Are there no T1 single cell C or N based growth rates for nitrate and phosphate? Alternatively, should the ammonium be calculated as a rate from T0 given that results for T1 are displayed, or is that for consistency to the other nutrient amendments?

Due to practical constraints and the intensiveness of single-cell measurements using SIMS, and had to prioritize samples for analyses. We specifically chose to measure the T1 sample in addition to T2 for the ammonium incubation, because NH₄ was rapidly taken up in our incubations.

We would like to clarify that the NH₄ uptake rates at T1 take into consideration the activity between tracer addition (T0) and the first biomass sampling time point after ca 6h (T1), while rates at T2 are calculated between tracer addition (T0) and the second biomass sampling time point after ca 30h (T2).

[241] So it does appear they are approaching Redfield ratio as the allocation of C and N are reconfigured in the cell. This is shown in Fig. 5, where the data, still with lots of

scatter, shows that at high nutrient additions, they opportunistically take up N into the cell, but that does not translate to higher C fixation. At low N availability, C-fixation drops to where many cells are close to Redfield. However, most cells lie above this line, so I agree with the authors interpretation.

We are happy that the reviewer agrees with our interpretation.

[311] The reference requires another bracket “[47]

Thank you, done.

[346] DOC retained on filters might require a reference, if that is known.

We now specify this to highlight that the DOC can be retained on the POM on the filters, instead of directly on the filters. Particularly exopolymeric substances can be very sticky (Nagar et al. 2021, *Extracellular polymeric substances in Antarctic environments: A review of their ecological roles and impact on glacier biogeochemical cycles*) and are visible around glacier surface biological material (McCutcheon et al. 2021, *Mineral phosphorus drives glacier algal blooms on the Greenland Ice Sheet*).

References:

Strzepek, R. F., Boyd, P. W., and Sunda, W. G., 2019, Photosynthetic adaptation to low iron, light, and temperature in Southern Ocean phytoplankton: Proceedings of the National Academy of Sciences, v. 116, p. 4388–4393.

Hopwood, M. J., Cantoni, C., Clarke, J. S., Cozzi, S., and Achterberg, E. P., 2017, The heterogeneous nature of Fe delivery from melting icebergs: Geochemical Perspectives Letters, v. 3, p. 200–209.

Reviewer #2 (Remarks to the Author):

The manuscript presents single-cell data on elemental composition and C and N assimilation of glacial ice algae, providing important new insights into their ecophysiology as well as the ecological and biogeochemical functioning of this system. The results are presented clearly, experiments seem mostly carefully conducted and discussion and conclusions adequate, however I have one concern regarding the experimental design that requires clarification and some questions/suggestions regarding interpretation (see below)

Specifically, based on the text in l. 420-421, and l. 426, do I understand it correctly that 1L of sample was spiked in a Schott bottle, then 750 ml were transferred to serum bottles for 30h incubations, while ca 250 ml remained in the Schott bottle for the 6h incubation? If so, it seems the conditions in 6h and 30h incubations were quite different (especially given the large headspace in the Schott bottle which, for instance, may have affected gas exchange, incl. DIC and ¹³C labelling). Couldn't this difference in incubation conditions explain the difference in C and N assimilation rates between t1 and t2?

The description by reviewer #2 of the experimental setup is correct. Note that the Schott bottles were kept at ambient light and temperature until the first samples could be taken (information added in lines 470-471). We do not think that the observed differences in rates are the result of methodological inconsistencies, as we took into consideration the potential difference in labeling% between incubation bottles by taking dedicated labeling% samples from both Schott (T1) and serum bottles (T2). We used the labeling% measured Schott bottles (T1) for all calculations of rates between T0 (tracer addition) and T1 (biomass/medium sampling from Schott bottles after 6h). For all calculations of rates

between T0 (tracer addition) and T2 (biomass/medium sampling from serum bottles after 30h), we used the average of labeling% from Schott Bottles (T1) and serum bottles (T2), as the cells experienced changing conditions during the incubation. For DIC, we observed no concentration differences (but slight difference in ^{13}C -DIC labeling%) between Schott and Serum bottle samples. As both ammonium and nitrate are dissolved in the liquid phase, we do not expect varying headspace volumes to affect the community activity.

Please also note that the C-based growth rate estimates are very similar for rates calculated between T0-T1 and T0-T2. There is however a difference between T0-T1 and T0-T2 N-based growth rates, which likely stems from the rapid uptake and depletion of the added ^{15}N tracer (leaving little N for assimilation between T1 and T2). We interpret this as the community efficiently scavenging bioavailable N sources, but being able to sustain consistently high C-based growth also in absence of additional N (lines 280-287 in discussion).

The authors conclude from the nutrient addition experiments that the algae are not nutrient limited (e.g. l. 386) and further that they can readily colonize new ice surfaces (ll. 390-394) – I guess nutrients are only part of the picture, doesn't this also depend on ecological interactions (i.e. how growth rates compare to those of potential competitors)? Also, at least in theory the lack of a response to the nutrient additions could indicate that some other (micro)nutrient (other than those tested) is limiting, can this be briefly discussed?

We agree with the reviewer that other factors may also play a role in limiting algal growth. While our focus has been on macronutrient limitation, we now acknowledge that micronutrients could limit algal growth if not available in sufficient concentrations. We have added a section addressing this point in the discussion (discussion: lines 247-253).

Curiously, the study shows lower growth rates in nutrient-amended bottles than in controls. As an explanation, in l. 301 it is suggested that these algal populations might be sensitive to high nutrient loads – what would be the mechanism here, could this be explained in a little more detail?

Thank you for addressing the missing mechanistic explanation. We can only speculate, as we did not study these responses on a metabolic level. Even though, the $\sim 10\ \mu\text{M}$ of added NH_4 concentration is below the threshold commonly considered toxic (most microalgae tolerate NH_4 concentrations $< 100\ \mu\text{M}$ (Collos and Harrison, 2014)) and $10\ \mu\text{M}$ NH_4 concentrations were also used in McCutcheon et al. (2021), we cannot exclude that the glacial microbes thriving at ultra oligotrophic conditions could be sensitive to high nutrient loads. We clarified that a high N load may negatively affect the algae metabolism (discussion, line 241).

Other minor comments:

Table 2 shows mean values for two time points but SD only for one of them. I suggest showing both time points with their respective SD.

From T1, we only have one biological replicate and thus no SD.

fig. S2b and fig. 1c are the same – I believe one could be omitted?

Thank you for spotting this. We removed Figure S2b.

Also, I think it would be helpful to indicate any chytrids in the microscope images by arrows. To non-specialists it is not clear whether the spherical 'cells' are algal cysts or chytrids

We marked the spheres that we identify as chytrids (sporangia) in Figure S2c-e.

I. 370 discusses the role of variable activity for adapting to environmental gradients over time - what about spatial (micro)environmental gradients in the ice, are those relevant?

Yes, the observed variability very likely plays a role in adaptation to spatial environmental gradients on the ice, and within the biofilm on the ice surface. We added this information to the revised manuscript (L 330-331).

Table 1 – state the units

Thank you, we added units to nutrient concentrations.

I. 99 typo – delete 'highlights the'

Thank you, done.

I. 339-341 revise phrasing, 'despite' does not seem to make sense

We revised the sentence accordingly.

I. 477-478 typo - delete 'again'

Thank you, done.

I. 776 journal name is missing

Thank you, we added the journal name.

I. 930, 931 please correct spelling of N₂ fixation

We corrected the mistake.

Supplement I. 24 'assumably' - presumably?

Thank you, done as suggested.